

# Evaluation of Arctic Ocean surface salinities from SMOS and two CMEMS reanalyses against in-situ datasets

Jiping Xie[1], Roshin P. Raj[1], Laurent Bertino[1, 2], Annette Samuelsen[1, 2], and Tsuyoshi Wakamatsu[1, 2]

1. Nansen Environmental and Remote Sensing Center, N5006 Bergen, Norway
2. Bjerknes Centre for Climate Research, Bergen, Norway

*Correspondence to*: Jiping. Xie (jiping.xie@nersc.no)


**Abstract**
Although the stratification of the upper Arctic Ocean is mostly salinity-driven, the sea
surface salinity (SSS) is still poorly known in the Arctic, due to its strong variability
and the sparseness of in-situ observations. Recently, two gridded SSS products have
been derived from the European Space Agency's (ESA) Soil Moisture and Ocean
Salinity (SMOS) mission, independently developed by the Barcelona Expert Centre
(BEC) in Spain and the Ocean Salinity Expertise Center (CECOS) of the Centre Aval
de Traitemenent des Donnees SMOS (CATDS) in France, respectively. In parallel,
there are two reanalysis products providing the Arctic SSS in the framework of the
Copernicus Marine Environment Monitoring Services (CMEMS), one global, and
another regional product.  While the regional Arctic TOPAZ4 system assimilates a
large set of sea-ice and ocean observations with an Ensemble Kalman Filter, the
global reanalysis combines in-situ and satellite data using a multivariate ensemble
optimal interpolation method. In this study, focused on the Arctic Ocean, these four
salinity products, together with the climatology both World Ocean Atlas (WOA) of
2013 and Polar science center Hydrographic Climatology (PHC), are evaluated
against in-situ datasets during 2011-2013. For the validation the in-situ observations
are divided in two; those that have been assimilated and those that have not. The
deviations of SSS between the different products and against the in-situ observations
show largest disagreements below the sea-ice and in the marginal ice zone (MIZ),
especially during the summer months. In the Beaufort Sea, the summer SSS from
the BEC product has the smallest - saline - bias (~0.6 psu) with the smallest root
mean squared difference (RSMD) of 2.6 psu. This suggests a potential value of
assimilating of this product into the forthcoming Arctic reanalyses.
**Keywords**: Arctic Ocean; sea surface salinity; SMOS; reanalysis; absolute deviation;

34


## 1. Introduction

The sea surface salinity (SSS) plays a key role to track hydrological processes in the global water cycle through precipitation, evaporation, runoff, and sea-ice thermodynamics (Vialard and Delecluse, 1998; de Boyer Montegut et al., 2004; Sumner and Belaineh, 2005; Vancoppenolle et al., 2009; Yu, 2011). SSS is known to impact the oceanic upper mixing significantly (Latif et al., 2000; Maes et al., 2006; Furue et al., 2018) and via its dominance on the surface layer density (Johnson et al, 2012) the SSS variability affects the thermohaline circulation in the northern North Atlantic (Reverdin et al., 1997). Using a coupled atmosphere-ocean model and an observed SSS climatology dataset, Mignot and Frankgnoul (2003) attributed the interannual variability of the Atlantic SSS to two factors: anomalous Ekman advection and the freshwater flux.

Increase in the freshwater content of the Arctic Ocean due to melting of glaciers and sea-ice (McPhee et al., 1998; Macdonald et al., 1999), a significant change in the global warming scenario, can leads to changes in the salinity distribution and fresh water pathways (Steele and Ermold, 2004; Morison et al., 2012). However, the freshwater flux is regarded as one of the least constrained parameters due to the small-scale features of river discharge, precipitation, and glacial/sea-ice melt (e.g., Tseng et al., 2016; Furue et al., 2018). In general, to avoid salinity drift in the models, the sea-surface freshwater flux is adjusted directly or by restoring SSS to its corresponding climatological value.

Monitoring SSS from space is crucial for understanding the global water cycle and the ocean dynamics, especially in the Arctic Ocean where our knowledge of the SSS variability is limited due to non-homogenous and sparse in-situ data. The European Space Agency's (ESA) Soil Moisture and Ocean Salinity (SMOS) satellite launched in November 2009, consists of the Microwave Imaging Radiometer using Aperture Synthesis (MIRIAS) instrument, a passive 2-D interferometric radiometer operating in L-band (1.4 GHz, 21 cm), to measure the brightness temperature (BT) emitted from the Earth (Font et al., 2010; Kerr et al., 2010). The L-band microwave is highly sensitive to water salinity, which influences the dielectric constants in the sea, and has less susceptible to atmospheric or vegetation-induced attenuation than higher frequency measurements (Mecklenburg et al., 2012). Since its operational phase started in May 2010, SMOS provides the longest SSS record from space over the global ocean, even compared with the National Aeronautics and Space


Administration's (NASA) Aquarius mission (between 2011 and 2015) and its follow-
up SMAP (Soil Moisture Active and Passive, since 2015).
Committed to provide global salinities averaged over 10-30 days with an accuracy of
0.1 psu for open ocean, ESA is responsible to interpreter the MIRAS data into
SMOS Level 1 (L1) and Level 2 (L2) data through a set of sequential processors
(Mecklenburg et al., 2012; ESA, 2017). In the L1 processing stage, the three relevant
products of L1A, L1B, and L1C are respectively corresponded to the calibrated
engineering visibility, the outputs of image reconstruction and multi-angular BT at the
top of atmosphere (TOA). Over oceans, Level 2 products (L2OS) are comprised of
three different ocean salinities, together with the BTs at TOA and on the sea surface,
distributed by ESA with swath-based format (e.g., SMOS Team, 2016; ESA, 2017).
Under the efforts at national agencies in France and Spain respectively, two Level 3
(L3) data products of SSS are freely available, which are independently developed by
the Ocean Salinity Expertise Center (CECOS) of the Centre Aval de Traitemenent
des Donnees SMOS (CATDS) at IFREMER and the Barcelona Expert Centre (BEC).
Few studies comprehensively investigate their quality uncertainties in the Arctic
Ocean at same time, although these two SMOS products have been successfully
used to resolve the local salinity front (D'Addezio et al., 2016) or to improve the
precipitation estimate (Supply et al., 2018).
In parallel to these monitoring activities from space, an ocean reanalysis or a
climatology dataset is a practical choice for public users to understand the Arctic
SSS. In recent studies regarding the Arctic Ocean salinity, Uotila et al. (2018)
focused on the stratification of the averaged salinities in the ten popular reanalyses,
where the seasonal cycle of monthly salinity in the layer of 0-100 m (Figure 12 of
Uotila et al., 2018) shows a considerable spread among these reanalyses. Note that
the full assessment of the Arctic SSS products has been hindered by extraordinarily
poor in-situ data coverage in the Arctic domain. With the accumulated SSS data from
the SMOS mission, it is now possible to evaluate the estimated salinity products from
different sources on a basin scale. In this study, we use two reanalysis products
available from the Copernicus Marine Environment Monitoring Service (CMEMS).
The first reanalysis (CMEMS product id: ARCTIC-REANALYSIS-PHYS-002-003) is
derived from the TOPAZ system (e.g., Xie et al., 2017), a coupled ocean and sea-ice
data assimilation system using Ensemble Kalman filter to assimilate the available
ocean and sea-ice observations from CMEMS. This reanalysis represents the Arctic



component in CMEMS providing daily and monthly reanalysis for Arctic domain since
in 1991. Another product (CMEMS product id:
MULTIOBS_GLO_PHY_REP_015_002) is derived from the combination of in-situ
data and satellite measurements including SMOS by a multivariate optimal
interpolation (MOI) technique (Droghei et al., 2018). The two CMEMS products
respectively represent classical ocean reanalysis products and optimally merged
observational data products.
In this paper, we assess the performance of the two CMEMS reanalysis products in
comparison to the two SMOS SSS products together with the two climatology
datasets: WOA13 (World Ocean Atlas of 2013; Zweng et al., 2013) and PHC (Polar
Science Center Hydrographic Climatology version 3.0; Steele et al., 2001). We
further extend the evaluation using available in-situ salinity observations during the
years of 2011-2013 from different data sources. The evaluation against the in-situ
data is also expected to shed light on the uncertainty of the SMOS products towards
the reliable Arctic SSS monitoring program, which also give useful information
needed for the assimilation of the SMOS SSS products into ocean
forecast/reanalysis systems in near future. The paper is organized as follows: In
Section 2, all the assessed SSS products and reference in-situ data are described.
The monthly means of SSS from these six products are intercompared, and the
monthly deviations referenced to the TOPAZ SSS are analyzed in in Section 3.
Section 4 illustrates the quantitative evaluations of the SSS products against the
reference in-situ data, which are divided into two sets of observations based on
whether the observations had been assimilated into TOPAZ or not. A summary of
this study is provided in Section 5.

**2. Data description**
*2.1 Sea surface salinity from SMOS*
The SSS retrieval from SMOS is subject to biases coming from various unphysical
contaminations such as the so-called land-sea contamination and the latitudinal
biases likely caused by the thermal drift of the instrument. Based on different
statistical approach, march-up criteria, and SMOS data filtering flags, the CECOS
and the BEC have independently developed a processing chain to produce the
relevant Level 3 SSS product on regular grids. The concerned two SSS products are
respectively named CEC and BEC hereafter in this study.





• *BEC product*
This product was developed in by BEC targeting high latitudes Oceans and in the
Arctic Ocean, available from http://cp34-bec.cmima.csie.es (last access: June 2018).
The BEC SSS product was generated from ESA L1B (v620) products (SMOS-BEC
Team, 2016), and accumulates the salinity data over 9 days with a spatial grid
resolution of 25 km for the period of 2011-2013. Using a non-Bayesian approach
systematic bias of the L1B salinity data is debiased against reference SSS
extrapolated from Argo float at 7.5 m depth, which are provided by the Coriolis data
center (www.coriolis.eu.org). For further processing detail, see Olmedo et al. (2016).
The bias corrected data are spatio-temporally interpolated to the L3 binned maps.
Then their anomaly is blended with WOA09 SSS climatology (Antonov et al., 2010)
using optimal interpolation with 300 km influence radius to produce the final L3
regularly gridded, daily SSS product (OA L3 SSS). The OA L3 SSS maps are served
daily on regular 25 km grids for an average period of 9 days.
• *CEC product*
The third version of LOCEAN SMOS SSS L3 maps (L3_DEBIAS_LOCEAN_v3) were
released by the CECOS of CATDS in July 2018. These SSS maps with 9 days
accumulation period at every 4 days are provided from 16[th] January 2010 to 25[th]
December 2017. These products, using Equal-Area Scalable Earth (EASA) Grid in
which pixels have a constant area and longitudes are equally spaced but not
latitudes, have a spatial resolution of 25km freely available on FTP: ftp.ifremer.fr (last
access: December 2018). Beginning from the ESA L1B products, the BTs are
reconstructed under apodization window and interpolation procedure (Vergely and
Boutin, 2017). Based on a semi-empirical ocean surface model developed internally,
three different forward models in the L2 processors are implemented for the SSS
retrieval and relevant geophysical parameters (SST, wind, etc.). Only one of these
three SSSs from the L2 processors are used as L2OS on an EASE grid, similar to
ESA L2OS (v622) products. Using the Bayesian retrieval approach (Kolodzejczyk et
al., 2016), the SMOS systematic errors in the vicinity of continents are migrated to
improve the product quality. Further, 'de-biasing' method (Boutin et al., 2018), an
improved technique to correct systematic biases, has been used in this version of the
CEC product, where the non-Gaussianity distribution of SSS is taken into account,
refining the latitudinal correction at high latitude, and preserving the naturally
seasonal variability of SSS.




### 2.2 Sea surface salinity from the two reanalyses in CMEMS

- *The Arctic reanalyses from TOPAZ*

TOPAZ uses the version 2.2 of Hybrid Coordinate Ocean Model (HYCOM,
Chassignet et al., 2003; Bertino and Lisæter, 2008) coupled with a simple
thermodynamic sea ice model (Drange and Simonsen, 1996). In the sea ice model,
the elastic-viscous-plastic rheology (Hunke and Dukowicz, 1997) was used to
describe the ice dynamics. The model domain covers the Arctic Ocean and the
northern Atlantic Ocean with a horizontal resolution of 12-16 km. Along the model
lateral boundaries, the temperature and salinity are relaxed to a combined
climatology data from PHC and WOA. Near the northern model boundary, a
barotropic inflow at the Bering Strait is imposed to involve the impact of Pacific water,
which varies seasonally as indicated by observations. Due to the poor knowledge on
the river discharge into the Arctic, a monthly climatology is calculated by the
precipitation from the ERA interim (Simmons et al, 2007) averaged over 20 years,
which was ingested to the Total Runoff Integrating Pathways (TRIP, Oki and Sud,
1998) hydrological model. In the model, the river discharges are treated as an
additional mass exchange by a negative salinity flux. Near the surface, to avoid the
salinity drift (Tseng et al., 2016; Furue et al., 2018), a weak relaxation to the
climatological SSS (30 days decay) is used as most of other ocean models adopted
to constrain the areas where the difference to climatology is less than 0.5 psu.
In order to obtain a reliable and dynamically consistent reanalysis in the Arctic
Ocean, the deterministic EnKF (DEnKF; Sakov and Oke, 2008) has been
implemented in TOPAZ with an ensemble of 100 model members which are driven
by 6-hourly perturbed atmosphere forcing from EAR interim. In the system, various
ocean and sea-ice observations (e.g., Xie et al., 2016, 2018) are assimilated into the
HYCOM model states to produce the Arctic ocean and sea-ice reanalysis. The full
evaluation for the TOPAZ SSS has been hindered by poor coverage of in-situ data
over the Arctic domain, although Xie et al. (2017) had comprehensively assessed the
TOPAZ reanalysis during 1991-2013 against various types of ocean and sea-ice
observations.  The related SSS product from this reanalysis is named TP4 here after.

- *SSS from the multivariable Optimal Interpolation dataset*

The CMEMS product of MULTIOBS_GLO_PHY_REP_015_002 (Verbrugge et al.,
2018) combines the SSS observations from in-situ and satellite data, using optimal



interpolation (OI, Buongiorno Nardelli et al., 2016) and covers the years of 1993-2017
at weekly interval. This product available from http://marine.copernicus.eu (last
access: 10th December 2018), provides the global SSS estimates on a 0.25° x 0.25°
regular grid. The main datasets used during the OI processing are as follow: 1) the
quality controlled in-situ data, COriolis dataset for Re-Analysis (CORA, Cabanes et
al., 2013) distributed through CMEMS (product id:
INSITU_GLO_TS_OR_REP_OBSERVATIONS_013_002_A/B); 2) the objectively
analyzed SSS and SST data generated from the CORA analysis system also
distributed by CMEMS, which has been upscaled to the final grid as the first guess
field for the multidimensional OI. 3) The SMOS L3 binned (L3bin) data reprocessed
by SMOS-BEC at 0.25° grid, which are built separately for descending and
ascending orbits and their composite; 4) The daily Reynolds L4 AVHRR_OI Global
blended SST product is used on a 0.25° grid. Over the same time period (2011-2013)
covered by the BEC SSS, the extracted SSS from this product are used in this study,
named MOI for simplification hereafter.

*2.3 Salinity near surface from in-situ data*

Against the two SMOS products from and the two CMEMS reanalyses, the SSS from
in-situ data are acquired here from three quality-controlled datasets. The first data
source is CORA from CMEMS (product id:
INSITU_GLO_TS_REP_OBSERVATIONS_013_001_b). Initially developed to supply
in-situ data in real time to French and European operational oceanography program
before 2010 under the French program Coriolis, CORA contains temperature and
salinity profiles from various in-situ data sources (Cabanes et al., 2013). Since 2013,
the CORA dataset has been updated every year by the collected profiles in the last
full year. They include all the Argo profiles, moorings, gliders, XBT, CTD, and XCTD
data. The latest version of the dataset, CORA5.1, covers the period of 1950-2016.
Note that the profiles from CORA5.1 have been used in the aforementioned
reanalysis systems for TP4 and MOI. Shown in Fig. 1a, the number of SSS
observations from CORA5.1 are 24249 over the domain north of 52°N during the
years of 2011-2013, and most of them are located in the northern Atlantic oceans.
The second in-situ data sources is the Beaufort Gyre Experiment Project (BGEP,
http://www.whoi.edu/website/beaufortgyre/background, last access: 14th December



2018). Aiming at monitoring the natural variabilities of the Beaufort Gyre in the
Canada Basin, BGEP is maintaining a set of observing system programs since 2003
and providing in-situ observations over the Beaufort Gyre in every summer. From the
BGEP, the valid SSS observations are depicted by the marks (anti-triangle, square,
and start) in the right panel of Fig.1. Last of all, we use in-situ data from GO-SHIP
(the Global Ocean Ship-based Hydrographic Investigations Program, Talley et al.
(2017)) under Climate Variability and Predictability Experiment (CLIVAR). Specifically,
SSS observations in the Beaufort Sea are extracted from CLIVAR/GO-SHIP data
with EXPOCODE (33HQ20111003 and 33HQ20121005, ref. Mathis and Monacci,
2014), which are available from https://cdiac.ess-
dive.lbl.gov/ftp/oceans/CARINA/Healy/ (last access: 18th December 2018). All the
valid salinity profiles are averaged within the upper 5 m layer near surface, in order to
obtain the marched observations of SSS for evaluation.

**3.  Intercomparison of monthly SSS**
Prior to the intercomparison of different SSS products, all the gridded products from
satellite, reanalysis and climatology have been converted on the same grids as used
in TP4 by nearest interpolation method. To quantitatively evaluate the SSS deviation
in the Arctic, the bias and the root mean square difference (RMSD) are defined by
$$\text{Bias} = \frac{1}{p}\sum_{i=1}^{p}(\mathbf{H_i}\mathbf{x_i^f} - \mathbf{s_i}) \qquad (1)$$

$$\text{RMSD} = \sqrt{\frac{1}{p}\sum_{i=1}^{p}(\mathbf{H_i}\mathbf{x_i^f} - \mathbf{s_i})^2} \qquad (2).$$

Where p is the evaluated times, $\mathbf{x_i^f}$ is the valid salinity from different sources at the $i$th
time, which is compared to the referred salinity field $\mathbf{s}_i$ and $H_i$ is the observation
operator if needs to project $\mathbf{x_i^f}$ into $\mathbf{s}_i$.
Figure 2 shows the monthly means of SSS in March and reveals considerable
differences in the two SMOS products. Notable differences are found in the Nordic
Seas, Barents Sea, and around Labrador Sea in Northern Atlantic Ocean. In general,
overall SSS maps from SMOS products are consistent with SSS of the two
reanalysis products and the two climatology products, although the BEC SSS tends
to be more saline than the CEC. It is noticeable that the location of sea-ice edge in
the two SMOS products marches well with that of the TP4 reanalysis (Fig. 2a, d).
Outside of the sea-ice covered region in the Arctic (represented by the 15% sea ice



concentration in Fig. 2) there is a good agreement between the subpolar SSS fields
of the two reanalyses and the climatologies. Over the sea-ice covered region, the
TP4 shows a gradual decrease from the sea-ice edge in the Nordic Seas with the
minima around the Beaufort Sea and the East Siberian Sea (ESS; Fig. 2b), being
consistent with the result in the PHC (Fig. 2c). The features mentioned above,
especially the minimal center in the Beaufort Sea, are missing in MOI and WOA (Fig.
2e, f). The MOI and the WOA also show commonly a potential artificial projection
issue around the North Pole.
As a contrast in summer, Fig. 3 shows the SSS fields in September respectively from
the SMOS products, the reanalyses and the climatologies. Considerable differences
in the two SMOS products are also found in Fig. 3 similar to that shown in Fig. 2. The
SSS field from CEC is relatively fresher then the BEC. In comparison to the
climatologies, the BEC SSS reproduces a much better representation of the surface
salinity in this region. As to the SSS from the reanalyses (TP4 and MOI) and the
climatologies (PHC and WOA), Fig. 3 shows a good agreement in the Northern
Atlantic Ocean. However, the discrepancies among them collectively emerge under
the sea-ice cover in the Arctic. Over the sea-ice covered Arctic region, the TP4 and
the PHC share common features. On the other hand, MOI and WOA do not portray
similar features and also show a projection issue around the North Pole.
Further, we quantify the differences between the TP4 and other SSS products.
Figure 4 shows the deviations of the monthly mean SSS in August from the five
products (BEC, PHC, CEC, MOI, and WOA), referred to the TP4. The two SMOS
products (Fig. 4a, c) show coherently negative deviations (~2 psu) along the sea-ice
edge in the marginal seas of the Beaufort Sea, the ESS, the Laptev Sea, and the
Kara Sea. Highlighted on the Arctic domain (>60°N), the SSS deviation of BEC in
August is about -0.5 psu with RMSD of 1.51 psu. Away from the sea-ice edge, the
deviation of BEC has a slight positive bias widely distributed in the Northern Atlantic
Ocean. For the CEC SSS, the averaged deviation is about -0.42 psu with RMSD
about 1.73 psu. Notably clear negative deviations appear in both BEC and CEC
products consistently along the sea-ice edge in the Beaufort Sea, the ESS, the
Laptev Sea and the Kara Sea. However, the deviations of two SMOS products in
August have clear differences over the north Atlantic and Arctic domain. While the
CEC has considerable negative deviations in the northern Atlantic with a minimum





over 1 psu located at the north of Denmark Strait, it has relatively strong positive
deviations near the coasts of the marginal seas around the Arctic.
The deviations in the northern Atlantic in MOI (Fig. 4d) and the two climatology
products are surprisingly small (Fig. 4b, e). However, over the sea-ice covered region
and its surrounding sea waters, the differences are rather significant. The PHC has a
relatively small negative deviation over the majority of the Arctic and north Atlantic
Oceans (Fig. 4b). However, around the sea-ice edge, the deviations are much larger.
On the other hand, MOI and WOA have strong positive deviations over the Eurasian
basin (> 1 psu), with respective RMSD of 4.21 and 3.29 psu in the whole Arctic
region.
In September (Fig. 5d, e), the SSS deviations of MOI and WOA still show an
anomalously large RMSD of 2.96 and 2.28 psu respectively. The averaged SSS
deviation of PHC (Fig. 5b) becomes slightly less than in August mainly due to the
positive deviations along the sea-ice edge in the marginal seas. Although the two
SMOS SSS products from SMOS have the smallest deviation among the five
products (Fig. 5a, c) with RMSD less than 1.5 psu, the CEC has surprisingly strong
positive deviation of 0.42 psu along the marginal and coastal seas in contrast to the
negative deviation over the same area in August (Fig. 4).
The mean and RMSD of monthly mean SSS deviations for the five products relative
to TP4, are averaged over the Arctic domain and their time series are plotted in Fig.
6. Among the five products, MOI appears the strongest seasonality with the values
more than 4 psu for its RMSD deviation during July and August and around 2 psu
during the winter months. The corresponding mean deviations of MOI are over -2 psu
during summer months and -0.5 psu during winter months. WOA has the second
largest seasonality with RMSD deviation more than 3 psu during summer and a
mean deviation of about -1.5 psu. This suggests the MOI SSS is quite close to the
WOA in the Arctic domain. As for PHC, the RMSD varies around 1.5 psu through the
year, and its mean deviation has a significant seasonality of the mean deviations
over -0.5 psu during summer and less than 0.5 psu during winter. The RMSD
deviations show relatively weak seasonality in the two SMOS SSS products. During
summer months, the RMSDs of both products are about 1.5 psu, while during winter
months the RMSDs of BEC and CEC vary respectively about 0.5 and 1.0 psu.
Throughout the whole year, the RMSDs of BEC are consistently smaller than that of



CEC. This indicates that the BEC SSS keeps consistency with that from TP4,
although the mean deviations of BEC show a slight negative bias.

**4.  Evaluation by in-situ observations**
Referred to Eqs. 1-2, the quantitative misfits of SSS products from the SMOS, the
reanalyses and the climatologies are calculated against the discrete in-situ
observations described in Section 2.3. For TP4 and BEC, the SSS evaluation is
conducted on the in-situ observing dates. For CEC and MOI, the corresponding
evaluation is made at the product date nearest backwards in time to the observing
dates. For PHC and WOA, the in-situ observations are sorted to monthly bin and
evaluated in each month. As shown in Fig. 1a, the SSS observations from CORA5.1
during the three years are distributed unevenly over the pan-Arctic area. Due to the
non-homogenous distribution of the observations, the evaluation of the gridded SSS
products against in-situ observations is limited to the observational-dense domains.
Here, we specifically focus our evaluation over the two domains: the northern Atlantic
Ocean during the entire period and the Beaufort Sea during summer seasons when
the surface is exposed owing to the sea ice melting.

*4.1 In the northern Atlantic Ocean and Nordic Seas*
In the northern Atlantic Ocean including the sub-regions from S4 to S7 (Fig. 1a),
23626 salinity observations are available for this evaluation, corresponding to more
than 97% of all valid observations over the Arctic domain from CORA5.1. Figure 7
shows the mean deviation of SSS for each product during the years of 2011-2013.
Over the northern Atlantic oceans including the Norwegian Sea and the Greenland
Sea, the considerable negative biases (<-0.16 psu) are shown in the products of
CEC, PHC and WOA (Fig. 7c, d, f). Among of them, the CEC shows significantly high
spatial variability. The SSS products of BEC, TP4 and MOI (Fig. a, b, e) have
relatively small bias (<0.08 psu), especially the MOI shows the minimal deviations in
most of this region.
If only comparison of the SSS between the BEC and the TP4, the latter has two
stronger positive biases appearing along the southern Norwegian coast and along
the Greenland west coast, although it has obviously smaller bias than the BEC in the
open seas. Against the Argo profiles from the Coriolis data center, SMOS-BEC Team
(2016) found the RMSDs of the BEC SSS in the Arctic (>50°N) are mostly less than



0.4 psu, but also showing the interannual variability like in the summer of 2012 the
RMSD close to 0.8 psu.  The RMSDs of the BEC SSS in the northern Atlantic Ocean
(S6 and S7 in Table 1) are less than 0.4 psu, but near the coast regions (S4 and S5
in Table 1) the RMSDs are over 1 psu. It further indicates the BEC quality has a
strong dependency on the locations.
Figure 8 shows the Root Mean Square (RMS) deviations of SSS for the all products
over the northern Atlantic Ocean and the Nordic Seas. Averaged in the local domain,
the maximal deviation among the six products can be found about 1.0 psu in the
CEC (Fig. 8d) in which high spatial variability is also profound. The minimal deviation
among them is found about 0.4 psu in the MOI (Fig. 8e), in which similar magnitude
of the RMSDs are distributed over the entire domain relatively evenly. The deviations
of PHC and WOA (Fig. 8c, f) also show relatively evenly distributions around the
average of 0.51 and 0.59 psu respectively. In case of the BEC (Fig. 8a), the
averaged RMS deviation about 0.57 psu is partly attributed to the strong deviations
along the southern Norwegian coast and near the sea-ice edge in the Greenland
Sea, which also are found in the CEC. Owing to these high RMSD values along the
coast and the ice edge, the RMSD of the BEC is obviously higher than that of about
0.4 psu evaluated by SMOS-BEC Team (2016).  As for TP4 (Fig. 8b), we can confirm
that the SSS near the coast also are subject to strong deviation. Despite the RMSD
deviation in the TP4 over the open sea is less than 0.3 psu, but the averaged
deviation in the entire domain reaches to 0.61 psu.
Around the core Arctic region (S0-S3 in Fig. 1a), the western Barents Sea (S3 in Fig.
1a) is the only sub domain where the in-situ data from CORA5.1 covers densely
having 509 SSS observations. We expect a high reliability in the estimation of SSS
uncertainty over this area. The RMSDs for BEC, TP4 and MOI are around 0.35 psu,
around 0.5 psu for the climatologies, and growing up to 1.36 psu for CEC (see Table
1). In contrast, the sea-ice covered regions of S0, S1, and S2 are monitored by
CORA5.1 quite sparsely with number of SSS observations 19, 36, and 59
respectively during the three years. Thus, relevance of the evaluated bias and RMSD
in these regions are questionable. Next, we evaluate the SSS products over the
Beafort Sea against in-situ data fully independent from CORA5.1 to avoid using the
salinity profiles have been assimilated in the TOPAZ reanalysis.

*4.2 In the summer of Beaufort Sea*



Over the Beaufort Sea during the summer months of 2011-2013, the independent in-
situ data are obtained from the BGEP and the CLIVAR both described in Section 2.3,
whose locations are marked in Fig. 1b. Evaluations of the six SSS products against
the in-situ data in the summer Beaufort Sea are plotted in Fig. 9. The SSS
observations from in-situ data range from 15 to 33 psu. The BEC SSS ranges from
24 to 31 psu with a bias of 0.65 psu and RMSD of 2.63 psu. On the same panel, the
TP4 ranges from 26 to 32 psu, with a bias of 2.73 psu and RMSD of 3.85 psu. The
linear regression coefficients for BEC and TP4 are 0.6 and 0.15 respectively. It is
found that the significant deviations of BEC and TP4 from the in-situ observations are
attributed to the particular four observations around (136.4°W, 70.5°N) collected on
15$^{th}$ August 2011 of which locations are marked in Fig. 1b by anti-triangles. They
become on the continental shelf near the estuary of Mackenzie River, where the
strong fresh water signature could be originated to river discharge.
For the climatologies, the PHC ranges from 25 to 31 psu, which is similar to that of
TP4, with a bias of 1.77 psu and RMSD of 3.13 psu. Compared to the TP4 deviation
at the Makenzie River basin, the deviations of the PHC are quite similar, but slightly
lower range.  This infers that the strong positive bias in the TP4 at these points
mostly originated the SSS relaxation in the TOPAZ model towards the PHC
climatology. In case of another climatology, the WOA ranges from 12 to 31 psu,
much wider than the range of PHC. This contributes the minimal bias of the WOA
about 0.02 psu among the six products, over the Beaufort Sea during all the
summers. However, it should be noticed that the range of in-situ observations
becomes much wider under 24 psu, which contributes a major source of the large
RMSD over 3.0 psu for both of PHC and WOA. It further suggests both climatology
products have a big representing uncertainty over the coastal fresh sea water (<24
psu) dominated region in the Arctic Ocean.
The CEC SSS ranges from 18 psu to 34 psu which is significantly wider than the
range of the BEC. The SSS bias of CEC is about 2.7 psu and its RMSD is about 3.9
psu. Again, the CEC deviations from the in-situ observations become wider in the
range where the SSS is less than 24 psu. For the MOI, the satellite and in-situ data
combined product, a negative bias is significant of more than 4 psu and the RMSD is
more than 7 psu. Contrast to other five SSS products, the anomalously fresh SSS
observed around (140°W,71°N) near the estuary of Mackenzie River are represented
by further fresher values of around 12 psu in the MOI.



In order to characterize dependencies of the bias for the six SSS products against
the in-situ data, their absolute biases are paired plotted as a function of observed
SSS in Fig. 10. In general, all products show considerable deviations by the maxima
reaching 8 to 14 psu. While the absolute misfits of the most of SSS products
monotonically increase towards lower salinity range, the bias of MOI shows its peak
around 20 psu shown in Fig. 10c. The fourth-order polynominal curve function,
$$F(S) = p_1 S^4 + p_2 S^3 + p_3 S^2 + p_4 S + p_5 \qquad (3)$$
is then fit to the absolute bias for each of the SSS products, where S represents the
in-situ salinity. The fitting coefficients from $p_1$ to $p_5$ for each product are listed in Table
2. The norm residuals printed on each panel of Fig. 10 clearly show that fitting for
MOI contains the largest uncertainty while the minimal norm residuals no more than
7 $psu^2$ are obtained for BEC and TP4. This suggests the derived fitting curves for
BEC and TP4 have credible skill in charactering its error distribution as a function of
the observed SSS. Both curves monotonically decrease towards the salinity greater
than 28 (30) psu for BEC (TP4) and increase slightly afterwards. The absolute bias in
TP4 is consistently larger than that in BEC. Although with lower amplitudes, the fitted
curves of PHC and WOA have the similar functional forms of TP4 and BEC. Their
relative relation of the fitted curves, PHC being consistently larger than WOA, is also
similar to that between TP4 and BEC.

**5.  Conclusions**
In order to understand the uncertainty of monitoring and reproduction of the Arctic
SSS in existing multi-source datasets, the two gridded SMOS SSS products (BEC
and CEC), two CMEMS reanalyzed products (TP4 and MOI), and two climatologies
(PHC and WOA) are first evaluated by intercomparison and secondly against in-situ
data during the years of 2011-2013. The monthly means of SMOS SSS (Fig. 2 and
Fig.3) clearly show the two SMOS products have equivalent data coverage in winter
months but obviously different in summer months due to the applied different BT
filtering flags. The salinity patterns from TP4 and PHC are considerably close to each
other, which is consistent to the fact that the SSS in the TOPAZ model is relaxed to
the PHC SSS at each time step. The monthly SSS patterns of MOI are clearly close
to that of WOA, and they both show some partial incompatibility near the North Pole
owing to the map projection (shown as in Fig. 2).





Relative to the TP4 SSS, the deviations of the four products (BEC, MOI, WOA and
PHC) show similar magnitude over the open waters, but the CEC shows an obviously
negative bias (<-1 psu) over the region extending from the Iceland towards the
western side of Ireland (Fig. 4, 5). This significant negative bias of the CEC should be
paid further attention in future evaluation studies about this SSS product. In general,
the most significant differences among the SSS deviations relative to the TP4 are
found under the Arctic sea-ice cover and in its surrounding marginal seas.
The BEC SSS in August and September (Fig. 4, 5) shows consistent negative
deviations along the sea-ice edge in the Beaufort Sea and the Chukchi Sea, but the
CEC along the ice edge shows the opposite deviations in these two months. This
indicates special attention is necessary for selecting a suitable SMOS SSS product to
be assimilated into an ocean and sea-ice forecasting system. The two SMOS
products would give rise to significantly different impacts to the concerned ocean
mixing so that the SSS quantitative evaluations of two products for optimal selection
or blending would be worthy of further studying.
Focusing the core Arctic domain (>60°N), the deviations of the five SSS products
relative to the TP4 show the diversely seasonal characteristics (Fig. 6). The MOI has
the largest seasonality in which the RMSD varies from over 1.5 psu in winter to over
4 psu in summer. The second largest seasonality can be found in the WOA with the
RMSD ranges from 1.5 psu to 3.5 psu. The RMSDs of CEC and PHC show similar
seasonality, but their mean deviations have opposite phases. The CEC has positive
bias (>0.5 psu) in September and October, and negative bias (<-0.5 psu) in February
and March while the PHC has negative deviation during the summer months (June-
October) and positive deviation during the winter months (December-April). Last of
all, the BEC SSS shows negative bias of less than 0.5 psu for all months, and its
RMSD has the smallest magnitude among the six SSS products, which ranges from
about 0.5 psu in winter months to about 1.5 psu in summer months. This concludes
that the BEC SSS has the most consistent pattern with the TP4 among all the
evaluated SSS products.
Against the in-situ data from CORA5.1 which have been used in the TP4 and the
MOI, the quantitative evaluations of the six SSS products have been investigated in
the northern Atlantic Ocean and the Nordic Seas, but in the sea-ice covered region
they are hindered by the sparse observations in the Arctic. In the northern Atlantic
Ocean domain, the MOI and the TP4 have relatively small misfits against in-situ data





(Fig. 7, 8). For two climatology datasets, the WOA and the PHC, both show
considerable negative bias (<-0.16 psu) and large RMSD (>0.5 psu). The CEC
shows the biggest RMSD (>1 psu) among all the six SSS products and mostly
negative bias (<-0.16 psu) with high spatial variability. Similar strong positive salinity
biases along the south-west Norwegian coast and along the south-west coast of
Greenland Island, are also found in the BEC but smaller than that in the TP4.
Highlighting in the Beaufort Sea, there are 193 valid SSS observations from BGEP
and CLIVAR, which have not been used in the TP4 and much denser than the
corresponding coverage in CORA5.1 (Fig. 1a). The linear regression against these
independent SSS observations suggests the BEC has the smallest RMSD of 2.63
psu with a positive bias of 0.65 psu, and the CEC has larger RMSD of about 3.9 psu
with a larger positive bias of 2.71 psu (Fig. 9). Equivalently, the TP4 also shows large
RMSD of about 3.85 psu with a large positive bias of 2.73 psu, but they are obviously
smaller than the corresponding misfits of the MOI which has the RMSD of 7.18 psu
with larger negative bias of -4.3 psu. As for the two climatologies, the WOA and the
PHC both have RMSD more than 3 psu but with significantly small bias in the WOA.
Overall, the large uncertainty found in a linear regression of all products is attributed
to large product-observation mismatch for in situ salinity data less than 24 psu, which
are observed over the continental shelf near the estuary of Mackenzie River.
In order to characterize the product-data misfits, the absolute deviations of all six
products against in-situ data, the 4th order polynomial function is fitted to the
deviation as a function of observed salinity (Fig.10). The absolute deviations of most
of the products except for MOI monotonically decrease as observed salinity increase.
The norm residuals for BEC and TP4 are the smallest of 6.28 and 6.88, respectively,
among all six products and the fitted curves give certain confidence in estimating size
of error in each SSS products. The fitted curve reaches its smallest value of about
0.5 psu at the in-situ salinities of 28 psu and 30 psu for BEC and TP4 respectively.
Both fitted curves for CEC and MOI have large norm residuals of 16.7 and 64.20
respectively. Note that special attention must be paid in if applying the MOI in the
Arctic Ocean due to its large negative bias and RMSD, although its smallest misfits
against CORA data in the northern Atlantic oceans among others.
Validation of the SSS products against TP4 product and in situ data conducted above
suggest certain benefit can be expected in assimilating the SMOS product like the
BEC, into the TOPAZ Arctic ocean analysis-forecast system. The knowledge on error



structure in the SSS products earned in this study will help us to reasonably estimate
the observation error for the SMOS product which is required by a data assimilation
system. Due to the poor spatial coverages of CORA in situ data in the Arctic Ocean,
the more data especially from the Arctic Ocean marginal seas should be compiled
from independent data source for validating the SMOS SSS products. The newest
SMOS product (Olmedo et al., 2018) that covers the years of 2010-2017 became
available recently. Validation of the SMOS SSS product for the longer period together
with the extended in situ data is under preparation now as the next step.

**Acknowledgement**
The authors acknowledge the support of CMEMS for the Arctic MFC. Grants of
computing time (nn2993k and nn9481k) and storage (ns2993k) from the Norwegian
Sigma2 infrastructures are gratefully acknowledged. The BEC SSS is produced by
the Barcelona Expert Centre (www.smos-bec.icm.csic.) mainly funded by the
Spanish National Program on Space. The CEC SSS is distributed by the Ocean
Salinity Expertise Center (CECOS) of CATDS at IFREMER, France.

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

| Region | Bias (psu) | | | | | | RMSD (psu) | | | | | |
|---|---|---|---|---|---|---|---|---|---|---|---|---|
| | BEC | CEC | TP4 | MOI | PHC | WOA | BEC | CEC | TP4 | MOI | PHC | WOA |
| S0 | - | - | -.97 | ***-.44*** | -1.12 | -.64 | - | - | 1.38 | ***.52*** | 1.43 | 1.42 |
| S1 | .56 | 2.80 | 2.80 | -2.1 | .75 | ***-.17*** | 4.95 | 3.78 | 4.11 | 4.39 | 3.78 | ***2.04*** |
| S2 | -1.42 | ***.65*** | .70 | -2.74 | -1.77 | -1.37 | ***1.81*** | 2.15 | 2.41 | 3.87 | 2.90 | 2.71 |
| S3 | ***.05*** | -.70 | -.15 | -.22 | -.31 | -.27 | ***.33*** | 1.36 | .35 | .37 | .52 | .46 |
| S4 | .05 | -.15 | .16 | .14 | ***.04*** | .08 | 1.28 | 1.52 | 1.29 | 1.27 | 1.32 | ***1.26*** |
| S5 | -.06 | .16 | .20 | ***.05*** | ***.05*** | .13 | 1.87 | 1.95 | 1.83 | 1.82 | 1.80 | ***1.77*** |
| S6 | .09 | .10 | ***0.0*** | -.01 | -.10 | .04 | .32 | .66 | .13 | ***.11*** | .29 | .16 |
| S7 | .15 | .45 | ***.03*** | -.04 | -.25 | ***-.03*** | .39 | .89 | .33 | ***.23*** | .44 | .27 |

**Table** 2. The fitting coefficients about the absolute deviations as a function of the in-situ SSS for the six products using a polynomial curve function by 4 order (as Eq. 3).

| Product | $F(p_1, p_2, p_3, p_4, p_5, s)$ | | | | | Norm residual $(r^2)$ | Samples of in situ |
|---|---|---|---|---|---|---|---|
| | $p_1(\times 10^{-3})$ | $p_2$ | $p_3$ | $p_4$ | $p_5$ | | |
| BEC | -0.162 | 0.0177 | -0.6604 | 9.409 | -34.7806 | 6.88 | 72 |
| CEC | -0.632 | 0.0542 | -1.687 | 22.158 | -96.720 | 16.70 | 111 |
| TP4 | 1.293 | -0.124 | 4.359 | -67.952 | 404.356 | 6.28 | 193 |
| MOI | -1.119 | 0.128 | -5.302 | 94.124 | -591.313 | 64.20 | 185 |
| PHC | 0.943 | -0.0867 | 2.938 | -44.118 | 256.0477 | 11.47 | 193 |
| WOA | -0.131 | 0.0122 | -0.414 | 5.713 | -21.22 | 28.64 | 193 |


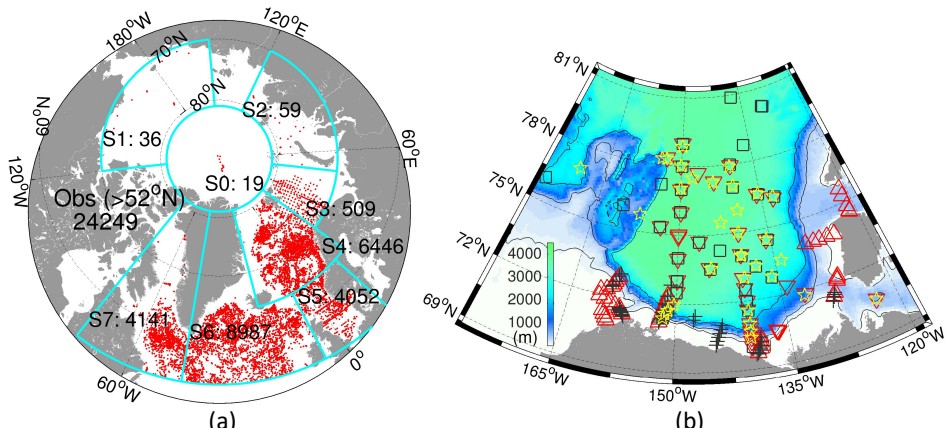

(a)                    (b)

**Fig**. 1 **(a)**: SSS locations of the in-situ observations north of 52°N in CORA5.1 during the years
of 2011-2013. They are divided into 8 regions around Arctic Ocean, and the number of
observations in each region are marked on the panel. **(b)**: SSS observations in the
Beaufort Sea during the summer months of 2011-2013. They are collected from the
BGEP (marked by anti-triangles, squares, and starts) and the CLIVAR (marked by
triangles and crosses) respectively, and with different color in which the red (black or
yellow) denotes the observations in 2011 (2012 or 2013).



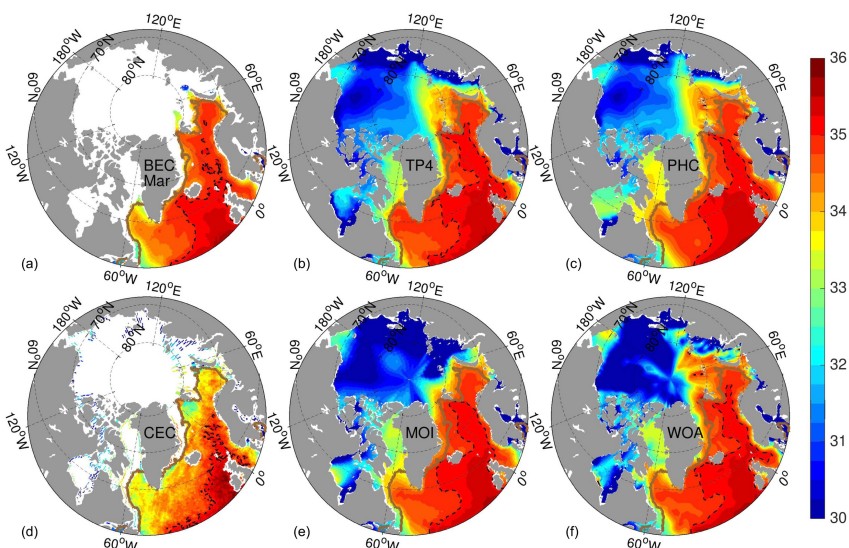

**Fig**. 2 Monthly SSS (unit: psu) in March from satellite products (BEC and CEC, *left column*), reanalyzes (TP4 and MOI, *middle column*), and climatology (PHC and WOA, *right column*). The thick brown line represents sea ice extent (15% concentration from TOPAZ4), and the black shaded isoline represents the salinity of 35 psu near surface.




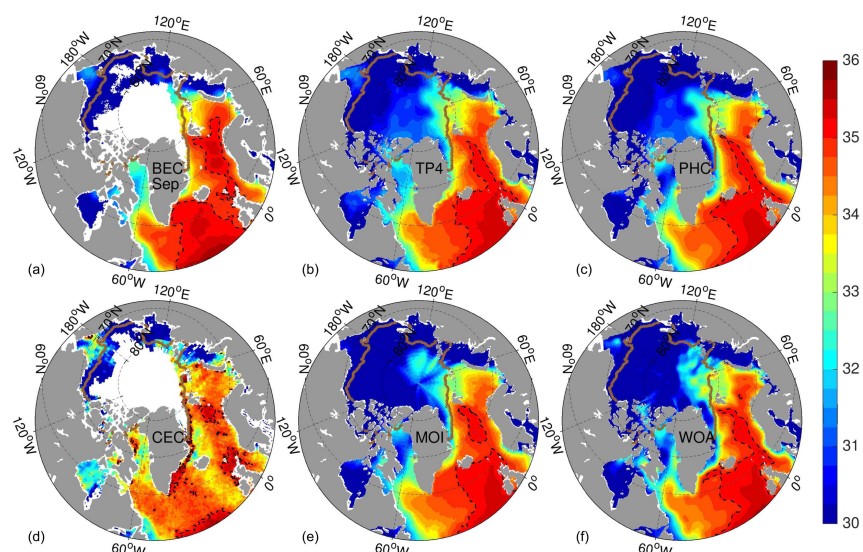

**Fig**. 3 Monthly mean of SSS (unit: psu) in September from satellite products (BEC and CEC, *left column*), reanalyzes (TP4 and MOI, *middle column*), and climatology (PHC and WOA, *right column*), other same as Fig. 2.




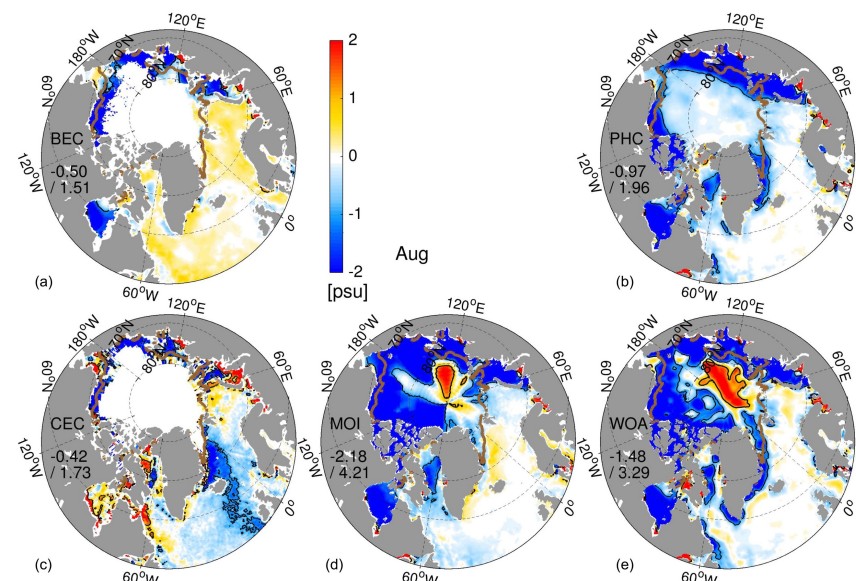

**Fig**. 4 Deviations of monthly SSS (unit: psu) in August for the 5 products of (a) BEC; (b) PHC; (c) CEC; (d) MOI; and (e) WOA relative to TP4. The thick brown line represents sea ice extent (15% concentration from TP4), the black lines represent ±1 psu.



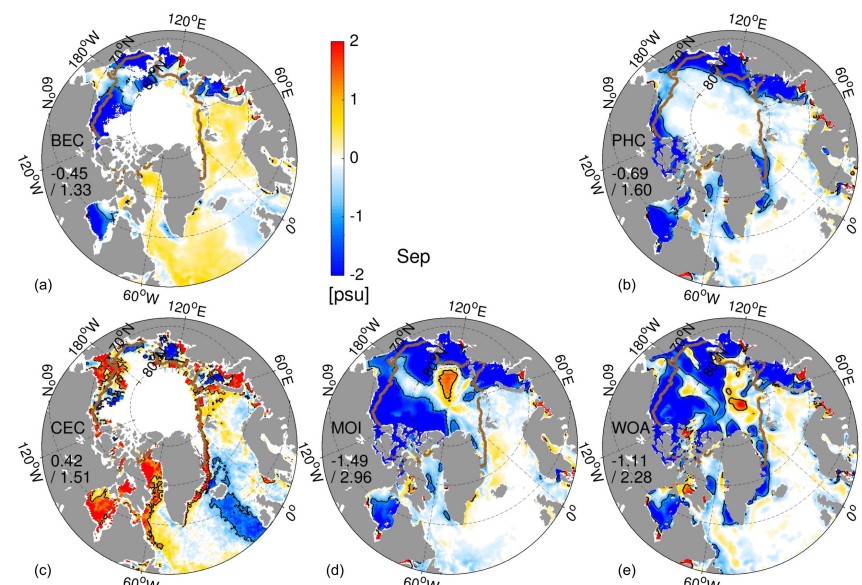

**Fig**. 5 Deviations of month SSS (unit: psu) in September for the 5 products of (a) BEC; (b) PHC; (c) CEC; (d) MOI; and (e) WOA relative to TP4. The thick brown line represents sea ice extent (15% concentration from TP4), the black lines represent ±1 psu.



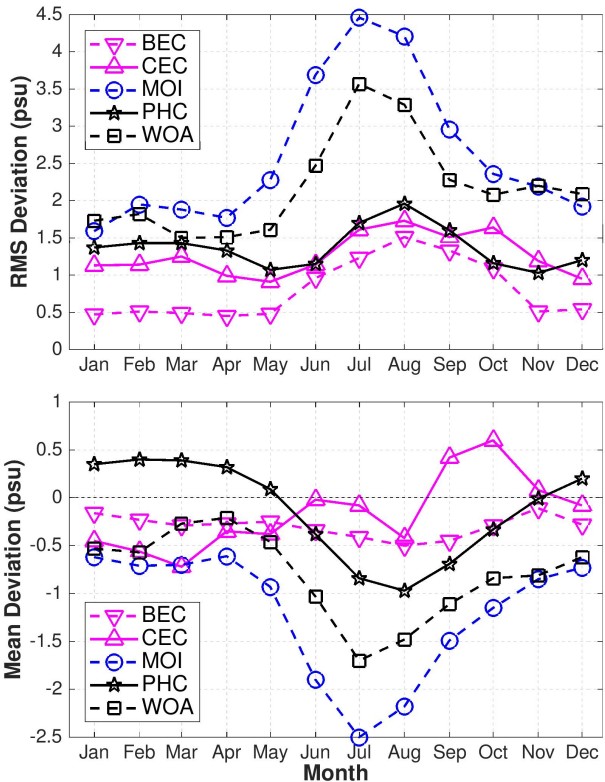

**Fig**. 6 RMSD (upper) and mean (bottom) deviations of monthly mean SSS (unit: psu) relative
to TP4 in the Arctic Ocean (>60°N) for the period of 2011-2013. The anti-triangle
(triangle, circle, star and square) line denotes the SSS deviations from BEC (CEC, MOI,
PHC and WOA) respectively.

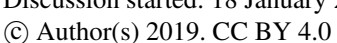



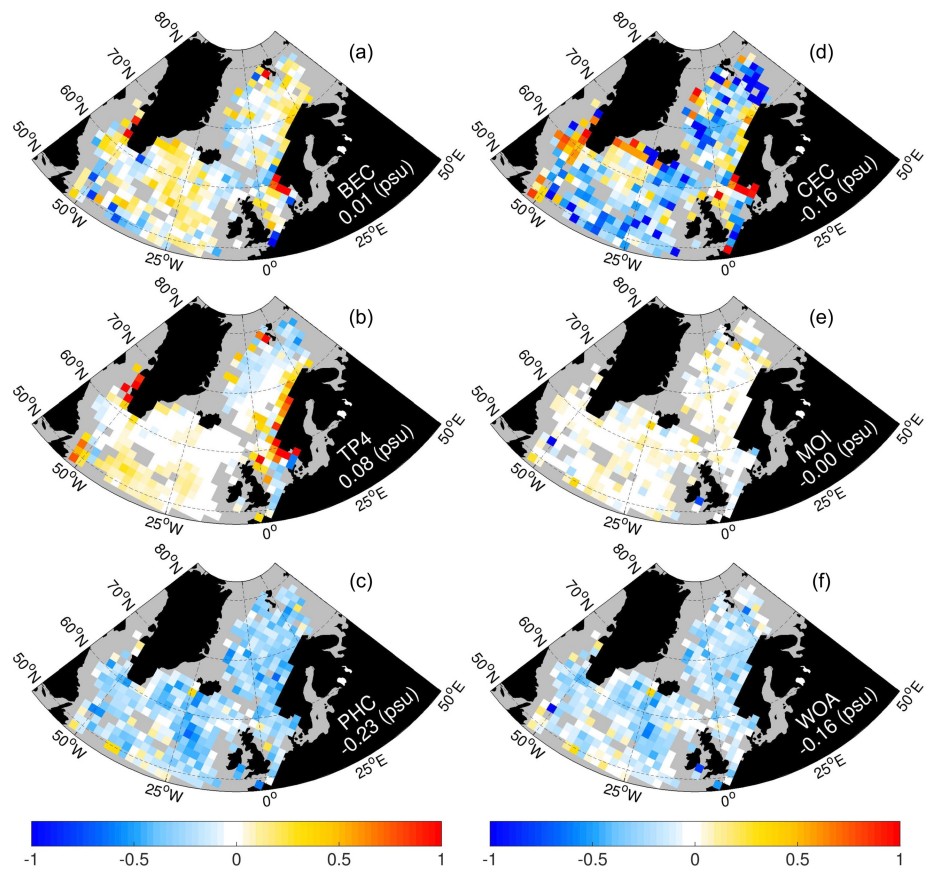

**Fig**. 7 The mean deviation of SSS for the six datasets compared to in situ observations from
CORA 5.1 during the three years of 2011-2013 in the northern Atlantic and Nordic seas.
The SSS observations are distributed into the coarse grid cells of 9x9 grids in TP4, with a
gray mask if the valid observations less than 10.




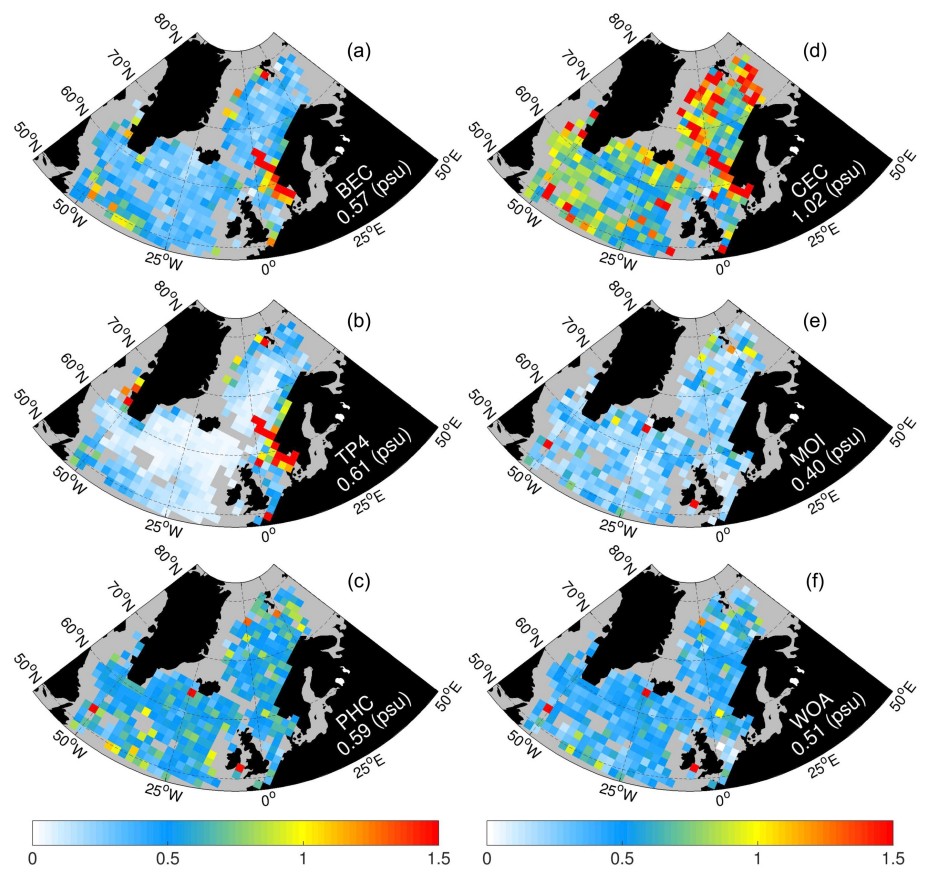

**Fig**. 8 The Root Mean Square deviation of SSS for six datasets compared to in situ observations
from CORA 5.1 during the three years of 2011-2013 in the northern Atlantic and Nordic
seas. The SSS observations are distributed into the coarse grid cells of 9x9 grids in TP4,
with a gray mask if the valid observations less than 10.





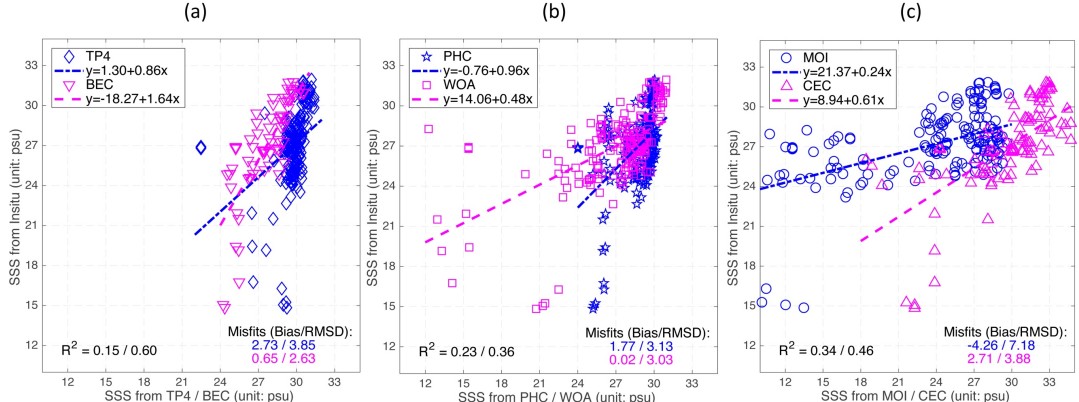

**Fig**. 9 Scatterplots of SSS compared to the in-situ observations in Beaufort Sea during the summer months of 2011-2013. Left: The diamond (anti-triangle) represents the SSS from TP4 (BEC) with blue (purple), and the linear regression is denoted by the dashed blue(pink) line. Middle: The star (square) from the climatology of PHC (WOA). Right: the circle (triangle) represents from MOI (CEC). The coefficient $R^2$ is the squared linear relationship, and the mean/RMS deviation also shown on the panels.

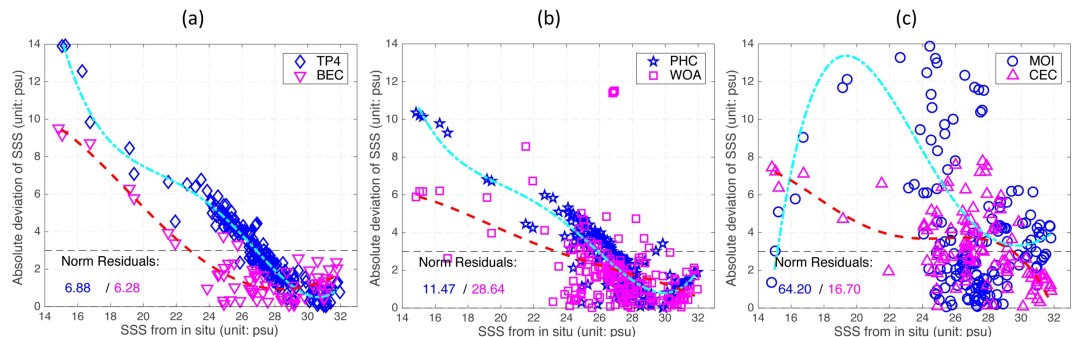

**Fig**. 10 Scatterplots of SSS uncertainty compared to the in-situ observations in Beaufort Sea as a function of the observed salinity. The black dashed line represents the absolute deviation of 3 psu. Left: The diamond (anti-triangle) represents from TP4 (BEC) with blue (purple). Middle: The star (square) from the climatology of PHC (WOA). Right: the circle (triangle) represents from MOI (CEC). The thick dashed curves are fitted by the fourth order polynomial function, and the norm residuals are marked on panel respectively.