# Peer review of "Evaluation of Arctic Ocean surface salinities from SMOS and two CMEMS reanalyses against in-situ datasets"

_Ocean Science, 2018_

## Referee Comment (RC1) · Anonymous Referee #1 · 12 Feb 2019

The paper shows an intercomparison among 6 arctic salinity products (2 based on SMOS acquisitions, 2 climatologies and 2 reanalysis products). All products are also compared with in-situ data CORA 5.1. In addition to the intercomparison by itself, the aim of the paper is to evaluate the best SMOS product to be assimilated by TOPAZ4 reanalysis product.

General Comments

The paper needs a general improvement of the writing. In some cases the concepts are no clear and English should be improved.

Other general comment refers to the version of the BEC SMOS product included in the

comparison. The version of the SMOS BEC product is only clearly defined at the end of the conclusions(lines 543-546). This should be explained in section 2. According to the expressed in conclusions, the version of the BEC SMOS product used in this paper is version 1.0. This version is not accessible now because it has been superseded by version 2.0. Why authors have not included v2.0 instead v1.0 in this study?

This reviewer knows the effort that implies to redo this validation using the new BEC product, but taking into account that v1.0 is not available, the inclusion of this product in the study is not interesting and v2.0 should be used. It is not necessary to proceed with all the period of the current v2.0, only the studied period (2011-2013) will be enough. Please, use v2.0 of 2011-2013 period instead v1.0. Change "BEC product" section accordingly.

Specific Comments

Lines 142-145: The BEC Arctic product 1.0 is not created as is described here. Systematic bias of the retrieved salinity data is corrected computing the so-called SMOS-climatology (the most probable value for a given lat-lon, incidence angle and across swath distance) and substituting this one by a reference. The used reference is the annual WOA13 (the same reference for all maps) and not Argo float extrapolated at 7.5 km. The second correction (the temporal bias correction) was computed for version 1.0 of the Arctic product in the same way as in the global one: assuming that the quantity of salt is constant in the surface. This coarse approach has been refined in version 2.0 (the current one) using Argo to compute the mean value of salinity for each Arctic map.

Line 147: The anomaly is referenced to WOA13 (not WOA09)

Section 3: Many comparisons are made involving different regions and products. A table similar to table 1 but for intercomparisons would help to the reader.

Line 466: Beware both SMOS products do not use different BT filtering flags. The main difference between both is that they are applying a completely different salinity retrieval

method.

Lines 5-9: Suppress "respectively". This long sentence probably sounds better as "Recently, two independent gridded SSS products have been derived from . . ..mission: the developed by the Barcelona. . . and the one developed by Ocean. . .." Here a mention about the regional or global character of both products will help to the reader to know about the general characteristics of each product (one can expect that a product specifically developed for Arctic will provide better results)

Line 42: "northern North Atlantic". Authors are referring to the north of the North Sea (a relative small region) or the authors are referring to the thermohaline circulation between Arctic Ocean and North Atlantic? (probably is this second option but "northern North Atlantic" sounds strange to me)

Lines 47-50. This sentence is difficult to read. "a significant change in the global warming scenario" should be " or a significant"? Probably no, but I do not clearly understand what is the meaning of this sentence.

Line 112: There exist, at least, two different versions of WOA13 (1.0 and 2.0) with significant differences between them in the Arctic data. Please, indicate the one used used in this study.

Line 130: "non geophysical sources" should be better than "unphysical contaminations"

Line 131: ice-sea contamination should be mentioned because is an important source of biases in the Arctic.

Line 193: Acronyms EnKF (Ensemble Kalman Filter) and DEnKF have not been defined in the text

Line 268: "Marches" should be "matches"?

Lines 281-282: Have in mind that comparison of BEC product and WOA is not recommended because BEC product incorporates as reference WOA13.

Lines 285-286: For this reviewer is not clear what do you mean with "over the sea-ice conver" and "under the sea-ice cover"... Under sea-ice cover means "below the ice"? Probably the meanin is related with latitudes not covered by ice?

Line 294: This sentence refers to figure 6? This figure is only referred in conclusions (line 487)

Line 413: The mentioned four observations, are outliers?

Line 536: In my opinion this is not a validation. Is a comparison.

Line 61. Typo: MIRIAS should be MIRAS

Technical corrections (Typos)

Line 122: Typo: "in in Section" ("in" written twice)

Line 133: Typo: "march-up" should be "match-up"?

Line 149: Correct address is http://bec.icm.csic.es

Line 166: Typo: should be EASE instead of EASA

Line 281: Typo: then should be than. The correct ending for the sentence should be "than the provided by BEC product"

Line 317:Word SMOS is used twice.

Line 552: The correct URL is bec.icm.csic.es

---

## Referee Comment (RC2) · Anonymous Referee #2 · 7 Mar 2019

The paper aims to quantify uncertainties of Arctic observation-based sea surface salinity to be included in the TOPAZ reanalysis. Two SMOS products are considered and compared against climatologies, observed data sets and reanalysis. This is an important problem in advancing in the data assimilation technics and improving the quality of CMEMS reanalyses. Anyway this study is not a significant step along that path. The paper has some unclear or incomplete reasoning. I do not feel that this research is ready to be published in OS. I do encourage resubmission after a much more detailed and careful investigation.

My primary concerns are i) the research is poorly presented, with vital details missing

ii) the BEC SMOS product selected from this study should actually be updated to version 2

iii) the PHC data set is old, is included in WOA13 and assimilated in TOPAZ. It does not add much to the analysis

iv) MOI is not a reanalysis. The CMEMS product MULTI-OBS_GLO_PHY_REP_015_002 is a combination of four data set. I do define a reanalysis as a combination of ocean modeling, data assimilation scheme and observed data sets. I would rather include in this study a global CMEMS ocean/sea ice reanalysis to be compared with TOPAZ4

v) The region of interested is the Arctic Ocean, but results are mostly related to the North Atlantic/Nordic Seas area

vi) Section 5 summarizes main results but a proper discussion to support the BEC SMOS and the "certain benefit (line 537) is missing.

These points significantly detract from the conclusions of the study, make the conclusions much weaker than the present manuscript states.

English need to be generally improved.
* * *

---

## Author Comment (AC1) · 12 Apr 2019

**Response to the comments from the anonymous Referee #1**

The paper shows an intercomparison among 6 arctic salinity products (2 based on SMOS acquisitions, 2 climatologies and 2 reanalysis products). All products are also compared with in-situ data CORA 5.1. In addition to the intercomparison by itself, the aim of the paper is to evaluate the best SMOS product to be assimilated by TOPAZ4 reanalysis product. -We thank the referee for the detailed evaluation of our manuscript and constructive suggestions. We appreciated this very much, the comments will all be taken into account in the revised version of the manuscript. Below, we answer point-by-point all specific and technical comments.

**General Comments**

The paper needs a general improvement of the writing. In some cases the concepts are no clear and English should be improved.

-We thank the reviewer for pointing to this weakness in the manuscript. We will improve the manuscript.

Other general comment refers to the version of the BEC SMOS product included in the comparison. The version of the SMOS BEC product is only clearly defined at the end of the conclusions (lines 543-546). This should be explained in section 2. According to the expressed in conclusions, the version of the BEC SMOS product used in this paper is version 1.0. This version is not accessible now because it has been superseded by version 2.0. Why authors have not included v2.0 instead v1.0 in this study?

-Thank you for this comment. We agree with the reviewer. Note that BEC product was released just before the deadline for the submission of the manuscript. The SMOS BEC product will be defined in Section 2.1. We will update all figures and results using version 2.0 in the revised manuscript.

This reviewer knows the effort that implies to redo this validation using the new BEC product, but taking into account that v1.0 is not available, the inclusion of this product in the study is not interesting and v2.0 should be used. It is not necessary to proceed with all the period of the current v2.0, only the studied period (2011-2013) will be enough. Please, use v2.0 of 2011-2013 period instead v1.0. Change "BEC product" section accordingly. -Agreed. The revision will replace the previous BEC product with version 2.0 and the concerned figures and analysis will be updated.

**Specific Comments**

Lines 142-145: The BEC Arctic product 1.0 is not created as is described here. Systematic bias of the retrieved salinity data is corrected computing the so-called SMOS climatology (the most probable value for a given lat-lon, incidence angle and across swath distance) and substituting this one by a reference. The used reference is the annual WOA13 (the same reference for all maps) and not Argo float extrapolated at 7.5km. The second correction (the temporal bias correction) was computed for version 1.0 of the Arctic product in the same way as in the global one: assuming that the quantity of salt is constant in the surface. This coarse approach has been refined in version 2.0 (the current one) using Argo to compute the mean value of salinity for each Arctic map.

-Thank you for this informative and constructive comment. The text will be changed to "The BEC SSS product was generated from ESA L1B (v620) products (SMOS-BEC Team, 2016), and accumulates salinity data over 9 days with a spatial grid

resolution of 25 km. To characterize the SMOS SSS bias and generate a timedependent bias correction, they use Argo salinity which is taken from the Argo profiles cut off at 10 m. Referred to the climatology of WOA13, the Argo profiles with anomalies larger than 5 psu in salinity or 10 in temperature were discarded (Olmedo et al., 2018)."

*Line 147: The anomaly is referenced to WOA13 (not WOA09)* -Thank you. It is corrected.

Section 3: Many comparisons are made involving different regions and products. A table similar to table 1 but for intercomparisons would help to the reader. -It is a good suggestion. A new table is added to clearly explain the reference data.

Line 466: Beware both SMOS products do not use different BT filtering flags. The main difference between both is that they are applying a completely different salinity retrieval method.

-Thanks for this point. It is corrected as "... show the two SMOS products have obviously different coverage in summer months due to the application of different methods of salinity retrieval."

Lines 5-9: Suppress "respectively". This long sentence probably sounds better as "Recently, two independent gridded SSS products have been derived from .... mission: the developed by the Barcelona... and the one developed by Ocean...." Here a mention about the regional or global character of both products will help to the reader to know about the general characteristics of each product (one can expect that a product specifically developed for Arctic will provide better results)

-Thank you for this suggestion. Here is the revised text: "Recently, two gridded SSS products have been derived from the European Space Agency's (ESA) Soil Moisture and Ocean Salinity (SMOS) mission: the one developed by the Barcelona Expert Centre (BEC) in Spain and another one developed by the Ocean Salinity Expertise Center (CECOS) of the Centre Aval de Traitemenent des Donnees SMOS (CATDS) in France. Except of their independent SSS retrieval algorithms, the former also developed a new version of regional grid products with respect to the Arctic Ocean, although both of them can cover global ocean."

Line 42: "northern North Atlantic". Authors are referring to the north of the North Sea (a relative mall region) or the authors are referring to the thermohaline circulation between Arctic Ocean and North Atlantic? (probably is this second option but "northern North Atlantic" sounds strange to me)

-Thank you for this comment. Here it means the thermohaline circulation between Arctic Ocean and North Atlantic. And this sentence will be changed as "the SSS further affects the decadal variability of hydrography in the upper North Atlantic (Reverdin et al., 1997)".

Lines 47-50. This sentence is difficult to read. "a significant change in the global warming scenario" should be "or a significant"? Probably no, but I do not clearly understand what is the meaning of this sentence.

-This sentence will be corrected by "On the other hand, the increased melting of glaciers and sea-ice in the Arctic (McPhee et al., 1998; Macdonald et al., 1999) leads

to significant changes in the salinity distribution and fresh water pathways (Steele and Ermold, 2004; Morison et al., 2012)."

- Macdonald, R. W., Carmack, E. C., McLaughlin, F. A., Falkner, K. K., and Swift, J. H.: Connections among ice, runoff and atmospheric forcing in the Beaufort Gyre. Geophys. Res. Lett., 26, 2223–2226, 1999
- McPhee, M. G., Stanton, T. P., Morison, J. H. and Martinson, D. G.: Freshening of the upper ocean in the Arctic: is perennial sea ice disappearing? Geophys. Res. Lett. 25, 1729–1732, 1998.
- Morison, J., Kwok, R., Peralta-Ferriz, C., Alkire, M., Rigor, I., Andersen, R., and Steele, M.: Changing arctic ocean freshwater pathways. Nature, 481:66–70, 2012.
- Steele, M. and W. Ermold (2004) Salinity Trends on the East Siberian Shelves, Geophysical Research Letters, Vol. 31, L24308, doi:10.1029/2004GL021302, 2004.

Line 112: There exist, at least, two different versions of WOA13 (1.0 and 2.0) with significant differences between them in the Arctic data. Please, indicate the one used in this study. -Thanks for this point. It is the WOA13 version 2.0, and will be clearly stated in the revision.

*Line 130: "non geophysical sources" should be better than "unphysical contaminations"* -Thank you for this suggestion. It is corrected.

*Line 131: ice-sea contamination should be mentioned because is an important source of biases in the Arctic.*

-Thank you for this suggestion. Add a more statement like "Especially in Arctic, resolving the edge between sea-ice and sea water still is a challenge. Ice-sea contamination same as land-sea contamination (Martín-Neira et al., 2016), in which the brightness temperature bias appears around the area covered sea-ice, may also be an important source of biases for the salinity retrieval in the Arctic Ocean."

Line 193: Acronyms EnKF (Ensemble Kalman Filter) and DEnKF have not been defined in the text

-It is corrected.

*Line 268: "Marches" should be "matches"?* -It is corrected.

Lines 281-282: Have in mind that comparison of BEC product and WOA is not recommended because BEC product incorporates as reference WOA13. -Thank you for pointing out this issue. In fact, even the BEC product has been incorporated as reference WOA13, the updated evaluation of the BEC version 2.0

still shows a bit far from the referred climatology as shown in Fig. A1

---

## Author Comment (AC2) · 12 Apr 2019

**Response to the comments from the anonymous Referee #2**

*The paper aims to quantify uncertainties of Arctic observation-based sea surface salinity to be included in the TOPAZ reanalysis. Two SMOS products are considered and compared against climatologies, observed data sets and reanalysis. This is an important problem in advancing in the data assimilation technics and improving the quality of CMEMS reanalyses. Anyway this study is not a significant step along that path. The paper has some unclear or incomplete reasoning. I do not feel that this research is ready to be published in OS. I do encourage resubmission after a much more detailed and careful investigation.*
-We thank the referee for the time spent and for the detailed revision of our manuscript. We appreciated very much for the comments which will be all taken into account in the revised version of the manuscript. Below, we answer point-by-point for all comments.

*My primary concerns are i) the research is poorly presented, with vital details missing*
-Thank you for this comment. We will improve the presentation of the work to help the reader understand. This evaluation has two parts: the first part is an intercomparison with reference to the TP4 reanalysis, the second part is an evaluation with respect to two in situ datasets which are involved in the TOPAZ system and independent respectively. In the revision, the observed SSS by in situ near surface will be extended from no deeper than 5 m depth to 8 m depth, which will involve more observation samples for this evaluation.

*ii) the BEC SMOS product selected from this study should actually be updated to version 2.*
-Yes. In the revision, we will use the version 2.0 of BEC product to replace all the figure and the concerned analysis (also see the response to the same comment of Referee #1).

*iii) the PHC data set is old, is included in WOA13 and assimilated in TOPAZ. It does not add much to the analysis*
-Thank you for this suggestion. The PHC dataset is one of the most important climatology in the Arctic Ocean, and still implemented widely in quantitative evaluation works (Carton et al., 2018, 2019). The PHC is based on the archive of observations primarily from the 1950s through the 1980s and so may have a somewhat cool climatology. In the current version of TOPAZ, the combined climatology of PHC and WOA13 are used as relaxation so that the quantitative comparison of two climatologies still could be helpful to reasonably reject this or not for the improving of the model relaxation process.

Carton, J.A., G.A. Chepurin, and L. Chen, 2018: SODA3: a new ocean climate reanalysis, *J. Clim.*, **31**, 6967-6983, doi:10.1175/JCLI-D-18-0149.1.

Carton, J.A., S.G. Penny, and E. Kalnay, 2019: Temperature and salinity variability in soda3, ECCO4r3, and ORAS5 ocean reanalyses, 1993-2015, *J. Clim.*, 32, 2277-2293, doi:10.1175/JCLI-D-18-0605.1.

*iv) MOI is not a reanalysis. The CMEMS product MULTIOBS_GLO_PHY_REP_015_002 is a combination of four data set. I do define a reanalysis as a combination of ocean modeling,*

*data assimilation scheme and observed data sets. I would rather include in this study a global CMEMS ocean/sea ice reanalysis to be compared with TOPAZ4.*

-Thank you for this comment. We agree that evaluating more global reanalysis products in CMEMS would be very interesting, and give more knowledge of the uncertainties in the different model systems, but it would go beyond the initial aim of directing our next assimilation work.

As an objective analysis product MOI uses the multivariable optimal interpolation method and can be used as a special representative in reanalysis products just like Simple Ocean Data Assimilation (SODA, Carton et al., 2018) is often used for comparative analysis with respect to other traditional reanalysis products (Uotila et al., 2018).

So in this study, we choose to use these two representative types of reanalysis products in CMEMS to evaluate the new satellite SSS products.

*v) The region of interested is the Arctic Ocean, but results are mostly related to the North Atlantic/Nordic Seas area.*

-Thank you for this comment. In the current evaluation, the comparison in the Beaufort Sea is presented in Fig. 9 and 10, referred to the independent SSS from BGEP and CLIVAR, which is directly linked to one of the main conclusions to support the BEC product. In addition, our forecast products more focus on the region north of 62N, where our general interest is and discussed in this study.

In this study, the in situ SSS from CORA5.1 were used by the TOPAZ system either assimilated or filtered during pre-processing for QC. It results these dependent SSS from CORA5.1 primarily are distributed in the Nordic Sea as shown in that figure. There are in general few observations in Arctic, the additional reason is a strictly used limit for the SSS observations - near the surface no deeper than 5 m depth. In fact, if extending the limit to 8 m depth, more SSS observations extracted from Ice-Tethered Profiler (ITP) will be involved. The Fig. A shows the locations of the SSS from ITP in the three years. Clearly, the evaluation referred to this dataset would enrich our knowledge of the Arctic SSS uncertainty.

[Figure]

**Fig**. A The locations of the SSS observations extracted at the 8m depth from the ITP living more than 30 days during the years of 2011-2013.

*vi) Section 5 summarizes main results but a proper discussion to support the BEC SMOS and the "certain benefit (line 537) is missing. These points significantly detract from the conclusions of the study, make the conclusions much weaker than the present manuscript states.*
-Thank you for this comment. The revision will add more discussions about this issue, with more consistence to the present results.

*English need to be generally improved.*
-Thanks for your comments. We will further improve the concerned parts.

---

## Author Response (AR1)

Dear Editor,

We thank the two reviewers for their critical and constructive comments on our research. Their comments have significantly improved our manuscript. The main modifications in the revision are las follows:

- Use the new version of the BEC SSS to replace the previously used in this study.
- Extend the SSS from 5 m to 8 m near the surface when extracted from insitu data, in order to involve the ITP observations.
- Rearrange the orders in Section 3 and Section 4 to highlight the evaluation against in-situ data, and which has been divided into two parts referred to dependent and independent in-situ dataset in the Sec.
  4.
- All figures have been updated due to the above two changes and two scatterplots showing the SSS against in-situ data from CORA5.1 are complemented in the revision.
- We have improved the English and expanded/shortened the text as recommended by the reviewers.

Below are the detailed responses to their comments: the reviewer comments are in black oud our response is in red.

**Anonymous Referee #1**

The paper shows an intercomparison among 6 arctic salinity products (2 based on SMOS acquisitions, 2 climatologies and 2 reanalysis products). All products are also compared with in-situ data CORA 5.1. In addition to the intercomparison by itself, the aim of the paper is to evaluate the best SMOS product to be assimilated by TOPAZ4 reanalysis product. -A: We thank the referee for the detailed evaluation of our manuscript and constructive suggestions. We appreciated this very much, all the comments are taken into account in the revised version.

**General Comments**

The paper needs a general improvement of the writing. In some cases the concepts are no clear and English should be improved.

-A: We thank the reviewer for pointing out this weakness in the manuscript. We have improved the English, in addition to extending, shortening and rearranging the text for improved clarity.

Other general comment refers to the version of the BEC SMOS product included in the comparison. The version of the SMOS BEC product is only clearly defined at the end of the conclusions (lines 543-546). This should be explained in section 2. According to the expressed in conclusions, the version of the BEC SMOS product used in this paper is version 1.0. This version is not accessible now because it has been superseded by version 2.0. Why authors have not included v2.0 instead v1.0 in this study?

-A: Thank you for this comment, we agree with the reviewer. Note that BEC product was released just before the submission of this manuscript. The SMOS BEC product used has been replaced by the version of 2.0, which is also defined in Section 2.1. We update all figures and results in the revised manuscript.

This reviewer knows the effort that implies to redo this validation using the new BEC product, but taking into account that v1.0 is not available, the inclusion of this product in the study is not interesting and v2.0 should be used. It is not necessary to proceed with all the period of the current v2.0, only the studied period (2011-2013) will be enough. Please, use v2.0 of 2011-2013 period instead v1.0. Change "BEC product" section accordingly. -A: Agreed. The revision will replace the previous BEC product with version 2.0 and the concerned figures and analysis are updated.

**Specific Comments**

Lines 142-145: The BEC Arctic product 1.0 is not created as is described here. Systematic bias of the retrieved salinity data is corrected computing the so-called SMOS climatology (the most probable value for a given lat-lon, incidence angle and across swath distance) and substituting this one by a reference. The used reference is the annual WOA13 (the same reference for all maps) and not Argo float extrapolated at 7.5km. The second correction (the temporal bias correction) was computed for version 1.0 of the Arctic product in the same way as in the global one: assuming that the quantity of salt is constant in the surface. This coarse approach has been refined in version 2.0 (the current one) using Argo to compute the mean value of salinity for each Arctic map.

-A: Thank you for this informative and constructive comment, the text is changed to:

Lines 135-143: "The BEC SSS product was generated from ESA L1B (v620) products, and accumulates salinity data over 9 days with a spatial grid resolution of 25 km. With respect to its previous version, a systematic bias in the retrieved salinity is corrected by computing the SMOS climatology (the most probable value for a given lat-lon, incidence angle and across-swath distance) which is substituted by a reference value from WOA13. In addition, a temporal bias correction has been refined in this version using near-surface Argo salinity to compute regional averages (see the details in Olmedo et al., 2018)."

*Line 147: The anomaly is referenced to WOA13 (not WOA09)* -A: Thank you. It is corrected.

Section 3: Many comparisons are made involving different regions and products. A table similar to table 1 but for intercomparisons would help to the reader.
-A: It is a good suggestion. A new table (Table 1) is added to clearly explain the product specifications. Moreover, the concerned sections are rearranged, and the evaluations against in-situ data are divide into two part according to dependent and independent observations in Section 4.

Line 466: Beware both SMOS products do not use different BT filtering flags. The main difference between both is that they are applying a completely different salinity retrieval method.

-A: Thanks for this point. It is deleted and replaced by other statement as Line 447: "... due to the different retrieval applied in these two datasets."

Lines 5-9: Suppress "respectively". This long sentence probably sounds better as "Recently, two independent gridded SSS products have been derived from .... mission: the developed by the Barcelona... and the one developed by Ocean...." Here a mention about the regional or global character of both products will help to the reader to know about the general characteristics of each product (one can expect that a product specifically developed for Arctic will provide better results)

-A: Thank you for this suggestion. Here is the revised text as

Lines of 2-6: "Recently two gridded Sea Surface Salinity (SSS) products that cover the Arctic Ocean have been derived from the European Space Agency's (ESA) Soil Moisture and Ocean Salinity (SMOS) mission: one developed by the Barcelona Expert Centre (BEC) and the other developed by the Ocean Salinity Expertise Center of the Centre Aval de Traitement des Données SMOS at IFREMER (CEC)."

Line 42: "northern North Atlantic". Authors are referring to the north of the North Sea (a relative mall region) or the authors are referring to the thermohaline circulation between Arctic Ocean and North Atlantic? (probably is this second option but "northern North Atlantic" sounds strange to me)

-A: Thank you for this comment. This sentence referred to the thermohaline circulation between Arctic Ocean and North Atlantic. And this sentence will be changed as

Lines of 41-43: "The SSS also affects the decadal variability of hydrography in the upper waters of the North Atlantic (Reverdin et al., 1997)."

Lines 47-50. This sentence is difficult to read. "a significant change in the global warming scenario" should be "or a significant"? Probably no, but I do not clearly understand what is the meaning of this sentence.

-A: This sentence will be corrected by the lines of 46-49:

"Additionally, the increased melting of glaciers and sea-ice in the Arctic (McPhee et al., 1998; Macdonald et al., 1999) leads to significant changes in the salinity distribution and fresh water pathways (Steele and Ermold, 2004; Morison et al., 2012)."

- Macdonald, R. W., Carmack, E. C., McLaughlin, F. A., Falkner, K. K., and Swift, J. H.: Connections among ice, runoff and atmospheric forcing in the Beaufort Gyre. Geophys. Res. Lett., 26, 2223–2226, 1999
- McPhee, M. G., Stanton, T. P., Morison, J. H. and Martinson, D. G.: Freshening of the upper ocean in the Arctic: is perennial sea ice disappearing? Geophys. Res. Lett. 25, 1729–1732, 1998.
- Morison, J., Kwok, R., Peralta-Ferriz, C., Alkire, M., Rigor, I., Andersen, R., and Steele, M.: Changing arctic ocean freshwater pathways. Nature, 481:66–70, 2012.

Steele, M. and W. Ermold (2004) Salinity Trends on the East Siberian Shelves, Geophysical Research Letters, Vol. 31, L24308, doi:10.1029/2004GL021302, 2004.

Line 112: There exist, at least, two different versions of WOA13 (1.0 and 2.0) with significant differences between them in the Arctic data. Please, indicate the one used in this study. -A: Thanks for this point. It is the WOA13 version 2.0, clearly stated in the revision.

*Line 130: "non geophysical sources" should be better than "unphysical contaminations"* -A: Thank you for this suggestion. It is corrected.

*Line 131: ice-sea contamination should be mentioned because is an important source of biases in the Arctic.*

-A: Thank you for this suggestion. The following statement has been added on 124-127:

"The SSS retrieval from SMOS is subject to biases originating from various nongeophysical sources such as the so-called land-sea contamination and the latitudinal biases, mainly caused by the thermal drift of the instrument. A particular challenge in the Arctic is the sea-ice edge because of ice-ocean contamination."

Line 193: Acronyms EnKF (Ensemble Kalman Filter) and DEnKF have not been defined in the text

-A: It is corrected.

*Line 268: "Marches" should be "matches"?* -A: It is corrected.

*Lines 281-282: Have in mind that comparison of BEC product and WOA is not recommended because BEC product incorporates as reference WOA13.*

-A: Thank you for pointing out this issue. In fact, even if the BEC product has incorporated WOA13as reference, the updated evaluation of the BEC version 2.0 still shows values far from the referred climatology as shown in Fig. A1. In the Barents

Sea, there is a clear salinity bias of BEC even if this product has been referred to WOA13.

Fig. A1 Monthly SSS (unit: psu) in March from satellite products of (a) BEC and (b) CEC, reanalyzes of (b) TP4 and (e) MOB, and climatology of (c) PHC and (f) WOA. The black shaded isoline represents the salinity of 35 psu near surface regarding to the product self.

The statement is changed as the Lines of 251-256:

"In comparison to the March situation, the BEC and CEC SSS in the Nordic Seas are both less saline, indicated by the 35 psu isoline. The sea ice masking of the two SMOS products differ considerably in the Canadian Basin and in the Arctic marginal seas. Although the SSS of TP4, MOB, PHC and WOA agree relatively well in the northern Atlantic Ocean, the discrepancies become dramatic in ice-covered areas."

Lines 285-286: For this reviewer is not clear what do you mean with "over the sea-ice conver" and "under the sea-ice cover"... Under sea-ice cover means "below the ice"? Probably the meanin is related with latitudes not covered by ice?

-A: Thank you for this point. It is corrected as the lines of 256-260:

"Below the ice or near the sea-ice edge (denoted by the brown thick line in Fig. 2 and 3), TP4 and PHC share common features, which can be explained by the model restoring to PHC. On the other hand, the MOB and WOA differ significantly in spite of WOA being used as input to the MOB."

*Line 294: This sentence refers to figure 6? This figure is only referred in conclusions (line 487)*

-A: No, it refers to Fig. 4 and Fig.6, this has been clarified in the text.

Line 413: The mentioned four observations, are outliers?

-A: Yes, a more detailed explanation had been included on lines 381-385:

"Looking at the low-salinity observations (~27 psu) collected at (136.4°W, 70.5°N) on 15th August 2011, marked by anti-triangles (Fig. 1b) near the Mackenzie River estuary, TP4 has a significant negative bias (

---

## Referee Report (RR1)

the authors did a satisfactory job in responding to my comments and rewriting their earlier draft.
Minor revisions are needed and English might be improved, still.

Line 24: rephrase "all six SSS products share a common challenge to represent fresh water masses"

Line 82: four SMOS products have been previously mentioned. Specify the two that are considered.

Line 88: add ocean to reanalysis products

Line 89: Uotila et al. presented temperature and salinity fields in the Arctic. Do you refer here to the seasonal cycle of both variables? Why the ten reanalysis are so different in the Arctic salinity, and all probably wrong? Topaz is part of the inter-comparison, add a more specific comment on results by Uotila et al.

Line 112: " can it also give…" is here the evaluation against in situ data, the subject?

Line 210: BGEP is available from CMEMS too, as required by the title of section 2.2?

Caption for Figure 1: only four sub-regions are in the Arctic Ocean, the others are located in the Nordic Seas and North Atlantic. Please add in the manuscript a clear definition of the Arctic domain, North Atlantic domain. The two are often mistaken in the text.

Line 243: the 35 psu isoline marks the Atlantic water that does only marginally reach the Arctic ocean. I suggest to add a lower-salinity isoline to the plot to better highlight also the inflow within the Arctic, something between 33 and 34 psu for example

Figure 2: the minimum salinity is not clearly shown, the blue saturates at 30 psu?
subplot e and f: are the salinity fields correct close to North Pole? It seems there is an issues in the interpolation at very high latitude for these two products

Line 252: The comparison is between BEC/CEC with all the other products, or BEC against CEC?

Line 255: I do not see that the 4 products agree in the North Atlantic. Rephrase

Line 260: what is exactly a universal reference?

Line 267: the Beaufort Sea is almost all ice-covered in CEC. The area that you consider here is unclear.

Line 268: CEC presents positive deviation in the Kara Sea close to coast line, probably due to the land-ocean interaction. Please add a line on that.

Line 273: I suggest to add a line on the missing low salinity related to the polar water that travels southward from the Arctic

Line 276: near and below the sea-ice cover reproduced by TP4?

Line 285: how is sea ice cover treated in all products in computing the deviations in Fig 6?
Line 293-294: is that evident in fig 6b?

Line 379: Rephrase. The range is larger, the salinity lower.

Line 447: rewrite "if it to be assimilated into"

---

## Author Response (AR2)

Dear Editor,

We thank the two reviewers for their critical and constructive comments on our research. Their comments have significantly improved our manuscript. The detailed responses to their comments are listed as following: the reviewer comments are in black and our response is in red.

**Anonymous Referee #1**

Although it is clarified in the text, it is not clear in figures 7, 8 and 11 the corresponding dataset for each provided value of R2 (R2=X/Y). Please, indicate the correspondence between X/Y and the datasets in the figure caption or by write X and Y numbers in a different color, according to each dataset, will help to the reader.

-A: Thanks for this comment. More explanation about R2 is added in the caption like "… R2 between the evaluated product and the in-situ SSS…" in Fig. 7, and same as in Fig.8 and Fig. 11.

**Anonymous Referee #2**

*The authors did a satisfactory job in responding to my comments and rewriting their earlier draft. Minor revisions are needed and English might be improved, still.*

-A: We thank the referee for the detailed evaluation of our manuscript and constructive suggestions. We appreciated this very much, all the comments are taken into account in the new revision. The English has been improved especially for the last parts of the paper.

*Line 24: rephrase "all six SSS products share a common challenge to represent fresh water masses"*

-A: Thank you for this comment. It is changed as

Line 21- 25: "When compared against independent in-situ data in the Beaufort Sea, the BEC product shows the smallest bias (<0.1 psu) in summer and the smallest RMSD (1.8 psu). The results also show that all six SSS products have a common challenge to represent fresh water masses (<24 psu) in the central Arctic."

*Line 82: four SMOS products have been previously mentioned. Specify the two that are considered.*

-A: Thank you for this comment. The more specific statement is added.

Line 81-83: "The present study thus investigates the accuracy of these two L3 SSS products from SMOS in the Arctic Ocean."

Line 88: add ocean to reanalysis products

-A: Thanks for this point, it is added.

Line 89: Uotila et al. presented temperature and salinity fields in the Arctic. Do you refer here to the seasonal cycle of both variables? Why the ten reanalysis are so different in the Arctic salinity, and all probably wrong? Topaz is part of the inter-comparison, add a more specific comment on results by Uotila et al.

-A: Thank you for this comment. Here, we only refer to the salinity seasonal cycle in Uotila et al. (2018). Although most reanalysis products (seven in the ten reanalyses in Table 1 of Uotila et al., 2018) restored salinity to climatology, it should be noticed that different salinity datasets were used, which also reveals the lack of a universal SSS reference. So we add the related comment in the text.

Line 90-93: "Although most reanalysis products (seven out of ten reanalyses in Table 1 of Uotila et al., 2018) restored salinity to climatology, they did not use the same salinity climatology, which betrays the lack of a universal SSS reference."

Line 112: " can it also give…" is here the evaluation against in situ data, the subject?

-A: Yes, it is. It is further corrected by "Can the evaluation against in-situ data also shed light on the uncertainties of the SMOS products?"

Line 210: BGEP is available from CMEMS too, as required by the title of section 2.2?

-A: Unfortunately not. This is why we list BGEP under Section 2.3, not 2.2. In this study, the in-situ observations from BGEP are directly downloaded from the website (http://www.whoi.edu/), and were not assimilated into TP4. The quantitative evaluation of SSS use that as one of the independent observations so we keep it in section 2.3.

Caption for Figure 1: only four sub-regions are in the Arctic Ocean, the others are located in the Nordic Seas and North Atlantic. Please add in the manuscript a clear

*definition of the Arctic domain, North Atlantic domain. The two are often mistaken in the text.*

-A: Thank you for this comment. In this study, the Arctic Ocean is limited to north of 60N. Here, considering the distributions of the valid in-situ observations from CORA5.1, the subregions are divided into 8 regions. Clearly, the subregions of S0-S4 are regarded as in the Arctic region, the other regions of S5-S7 are attributed into the northern North Atlantic.

So the caption for Fig. 1 has a change as "8 sub-regions divide the Arctic Ocean (S0-S4) and the northern North Atlantic Ocean (S5-S7), …"

In additional, more statement about this issue is added in Section 4.1
Line 322-326: "In this study, the Arctic domain (>60N) is the core region for evaluation, divided into five sub-regions numbered from S0 to S4. It contains the central Arctic (sub-regions S0, S1, S2, and S3) and the Nordic Seas (S4). The regions from S5 to S7 are in the northern North Atlantic."

*Line 243: the 35 psu isoline marks the Atlantic water that does only marginally reach the Arctic ocean. I suggest to add a lower-salinity isoline to the plot to better highlight also the inflow within the Arctic, something between 33 and 34 psu for example*

-A: Thank you this nice suggestion. We add the isoline of 33.6 psu and tuning the colorbar with a larger range as shown in the updated Fig. 2 and 3.

*Figure 2: the minimum salinity is not clearly shown, the blue saturates at 30 psu? subplot e and f: are the salinity fields correct close to North Pole? It seems there is an issues in the interpolation at very high latitude for these two products*

-A: Yes, the minimum salinity is not clearly shown in Fig.2 due to the colorbar is cut off when the salinity below 30 psu. So the colorbars in the new figures have been extended to represent fresh waters. In the central Arctic, the lower SSS in TP4 and PHC is around 30 psu, which is rather saline compared to that in MOB and WOA. Both suffer from interpolation artefacts due to their unfortunate regular lat-lon projection (singularity at the North Pole).

*Line 252: The comparison is between BEC/CEC with all the other products, or BEC against CEC?*

-A: The comparison is between BEC/CEC with all the other products, especially indicated by the dashed line of 35 psu in Fig. 3, they are both less saline.

*Line 255: I do not see that the 4 products agree in the North Atlantic. Rephrase*

-A: Thank you for this comment. The 4 products show the similar patterns by the dashed line (35 psu) in the North Atlantic and the Nordic seas. To avoid the misunderstanding, the text is changed at Line 257-259: "Although the SSS of TP4, MOB, PHC and WOA agree relatively well in the North Atlantic Ocean and the Nordic seas **as shown by the dashed lines of 35 psu**, …"

*Line 260: what is exactly a universal reference?*

-A: Here a universal reference means a common reference to Arctic SSS analysis that can be consensually accepted or used in both spatial and temporal resolution and accuracy.

*Line 267: the Beaufort Sea is almost all ice-covered in CEC. The area that you consider here is unclear.*

-A: Thank you for this comment. In August, the CEC SSS appears a smaller area than BEC in the Beaufort Sea. For the two SMOS products, we only consider ice free pixels.

*Line 268: CEC presents positive deviation in the Kara Sea close to coast line, probably due to the land-ocean interaction. Please add a line on that.*

-A: In fact, we noticed the positive deviations of CEC near the coast line (not only in the Kara Sea) which are rather significant even compared with that in BEC.
A concerned comment is added as
Line 272-273: "A positive deviation of CEC is noticeable in the Kara Sea, which indicates the land-ocean interaction stronger than that in BEC."

*Line 273: I suggest to add a line on the missing low salinity related to the polar water that travels southward from the Arctic*

-A: Thank you for this suggestion. A comment is added as

Line 280-284:"For the BEC and CEC products that use different ice masks, the deviations are averaged outside their respective ice mask, not their intersection. Comparing the low salinity lines of 33.6 psu in Fig. 3a and 3d, it clearly shows the polar water southward from Arctic has a misinterpretation owing to the used ice mask."

*Line 276: near and below the sea-ice cover reproduced by TP4?*
-A: Yes, thank you this remind. This definition is added as Line 285-286: "Near and below the sea-ice cover reproduced by TP4 (the thick brown line in the figures), …"

*Line 285: how is sea ice cover treated in all products in computing the deviations in Fig 6?*
-A: Figure 6 reveals the monthly deviations of the five SSS products referred to TP4, which is constrained at north of 60N without considering sea ice cover, although the two SMOS products only use ice free pixels. If averaging the deviations outside of ice cover (defined by 0.15 concentration in TP4), the monthly deviations of the five products referred to TP4 are shown in Fig. A as bellow. Clearly, the BEC and the CEC have similar deviation features like in Fig. 6, compared with other products except of the specific values.

[Figure]

Fig. A Monthly deviations in the Arctic Ocean (>60N; out of ice cover defined by TP4) of (a) the RMS and (b) the spatial average during the period 2011-2013 for the five SSS products referred to TP4. The anti-triangle (triangle, circle, star and square) line represents the SSS deviations from BEC (CEC, MOB, PHC and WOA respectively).

*Line 293-294: is that evident in fig 6b?*

-A: Referred to the TP4 SSS, the RMS deviation (Fig. 6a) of BEC has consistently smaller RMS compared with the other products. For the mean deviation (Fig. 6b), the same conclusion is evident for BEC except in the summer months. In summer, the SSS deviation of CEC clearly shows large deviations of opposite signs in Fig. 5c, which sums up to the smaller deviation compared to that in BEC in Fig. 6b.

*Line 379: Rephrase. The range is larger, the salinity lower.*

-A: Thank you for this point. It is revised as Line 392-393: "On the other hand, the range of TP4 SSS increases from 19 to 32 psu, with a larger saline bias of 2.59 psu and a RMSD of 3.63 psu."

*Line 447: rewrite "if it to be assimilated into"*

[revised manuscript text omitted]

Font color: Text 1

| Page 14: [1] Formatted | Authors | 8/13/19 9:35:00 AM |
|---|---|---|

Font color: Text 1

| Page 14: [2] Formatted | Authors | 8/13/19 9:35:00 AM |
|---|---|---|

Font color: Text 1

| Page 14: [2] Formatted | Authors | 8/13/19 9:35:00 AM |
|---|---|---|

Font color: Text 1

| Page 14: [3] Formatted | Authors | 8/13/19 9:35:00 AM |
|---|---|---|

Font color: Text 1

| Page 14: [3] Formatted | Authors | 8/13/19 9:35:00 AM |
|---|---|---|

Font color: Text 1

| Page 14: [4] Formatted | Authors | 8/13/19 9:35:00 AM |
|---|---|---|

Font color: Text 1

| Page 14: [4] Formatted | Authors | 8/13/19 9:35:00 AM |
|---|---|---|

Font color: Text 1

| Page 14: [5] Formatted | Authors | 8/13/19 9:35:00 AM |
|---|---|---|

Font color: Text 1

| Page 14: [5] Formatted | Authors | 8/13/19 9:35:00 AM |
|---|---|---|

Font color: Text 1

| Page 14: [5] Formatted | Authors | 8/13/19 9:35:00 AM |
|---|---|---|

Font color: Text 1

| Page 14: [5] Formatted | Authors | 8/13/19 9:35:00 AM |
|---|---|---|

Font color: Text 1

| Page 14: [5] Formatted | Authors | 8/13/19 9:35:00 AM |
|---|---|---|

Font color: Text 1

| Page 14: [5] Formatted | Authors | 8/13/19 9:35:00 AM |
|---|---|---|

Font color: Text 1

| Page 14: [5] Formatted | Authors | 8/13/19 9:35:00 AM |
|---|---|---|

Font color: Text 1

| Page 14: [5] Formatted | Authors | 8/13/19 9:35:00 AM |
|---|---|---|

Font color: Text 1

| Page 14: [5] Formatted | Authors | 8/13/19 9:35:00 AM |
|---|---|---|

Font color: Text 1

| Page 14: [5] Formatted | Authors | 8/13/19 9:35:00 AM |
|---|---|---|

Font color: Text 1

| Page 14: [5] Formatted | Authors | 8/13/19 9:35:00 AM |
|---|---|---|

Font color: Text 1

| Page 14: [5] Formatted | Authors | 8/13/19 9:35:00 AM |
|---|---|---|

Font color: Text 1

| Page 14: [5] Formatted | Authors | 8/13/19 9:35:00 AM |
|---|---|---|

Font color: Text 1

| Page 14: [5] Formatted | Authors | 8/13/19 9:35:00 AM |
|---|---|---|

Font color: Text 1

| Page 14: [5] Formatted | Authors | 8/13/19 9:35:00 AM |
|---|---|---|

Font color: Text 1

| Page 14: [5] Formatted | Authors | 8/13/19 9:35:00 AM |
|---|---|---|

Font color: Text 1

| Page 14: [5] Formatted | Authors | 8/13/19 9:35:00 AM |
|---|---|---|

Font color: Text 1

| Page 14: [5] Formatted | Authors | 8/13/19 9:35:00 AM |
|---|---|---|

Font color: Text 1

| Page 14: [5] Formatted | Authors | 8/13/19 9:35:00 AM |
|---|---|---|

Font color: Text 1

| Page 14: [5] Formatted | Authors | 8/13/19 9:35:00 AM |
|---|---|---|

Font color: Text 1

| Page 14: [5] Formatted | Authors | 8/13/19 9:35:00 AM |
|---|---|---|

Font color: Text 1

| Page 14: [5] Formatted | Authors | 8/13/19 9:35:00 AM |
|---|---|---|

Font color: Text 1

| Page 14: [6] Formatted | Authors | 8/13/19 9:35:00 AM |
|---|---|---|

Font color: Text 1

| Page 14: [6] Formatted | Authors | 8/13/19 9:35:00 AM |
|---|---|---|

Font color: Text 1

| Page 14: [6] Formatted | Authors | 8/13/19 9:35:00 AM |
|---|---|---|

Font color: Text 1

| Page 15: [7] Formatted | Authors | 8/13/19 9:35:00 AM |
|---|---|---|

Font color: Text 1

| Page 15: [8] Formatted | Authors | 8/13/19 9:35:00 AM |
|---|---|---|

Font color: Text 1

| Page 15: [9] Formatted | Authors | 8/13/19 9:35:00 AM |
|---|---|---|

Font color: Text 1

| Page 15: [10] Formatted | Authors | 8/13/19 9:35:00 AM |
|---|---|---|

Font color: Text 1

| Page 15: [11] Formatted | Authors | 8/13/19 9:35:00 AM |
|---|---|---|

Font color: Text 1

| Page 15: [12] Formatted | Authors | 8/13/19 9:35:00 AM |
|---|---|---|

Font color: Text 1

| Page 15: [13] Formatted | Authors | 8/13/19 9:35:00 AM |
|---|---|---|

Font color: Text 1

| Page 15: [14] Formatted | Authors | 8/13/19 9:35:00 AM |
|---|---|---|

Font color: Text 1

| Page 15: [15] Formatted | Authors | 8/13/19 9:35:00 AM |
|---|---|---|

Font color: Text 1

| Page 15: [16] Formatted | Authors | 8/13/19 9:35:00 AM |
|---|---|---|

Font color: Text 1

| Page 15: [17] Formatted | Authors | 8/13/19 9:35:00 AM |
|---|---|---|

Font color: Text 1

| Page 15: [18] Formatted | Authors | 8/13/19 9:35:00 AM |
|---|---|---|

Font color: Text 1

| Page 15: [19] Formatted | Authors | 8/13/19 9:35:00 AM |
|---|---|---|

Font color: Text 1

| Page 15: [20] Formatted | Authors | 8/13/19 9:35:00 AM |
|---|---|---|

Font color: Text 1

| Page 15: [21] Formatted | Authors | 8/13/19 9:35:00 AM |
|---|---|---|

Font color: Text 1

| Page 15: [21] Formatted | Authors | 8/13/19 9:35:00 AM |
|---|---|---|

Font color: Text 1

| Page 15: [22] Deleted | Authors | 8/13/19 9:35:00 AM |
|---|---|---|

▼

| Page 15: [22] Deleted | Authors | 8/13/19 9:35:00 AM |
|---|---|---|

▼

| Page 15: [23] Formatted | Authors | 8/13/19 9:35:00 AM |
|---|---|---|

Font color: Text 1

| Page 15: [24] Formatted | Authors | 8/13/19 9:35:00 AM |
|---|---|---|

Font color: Text 1

| Page 15: [24] Formatted | Authors | 8/13/19 9:35:00 AM |
|---|---|---|

Font color: Text 1

| Page 15: [24] Formatted | Authors | 8/13/19 9:35:00 AM |
|---|---|---|

Font color: Text 1

| Page 15: [25] Formatted | Authors | 8/13/19 9:35:00 AM |
|---|---|---|

Font color: Text 1

| Page 15: [26] Formatted | Authors | 8/13/19 9:35:00 AM |
|---|---|---|

Font color: Text 1

| Page 15: [27] Formatted | Authors | 8/13/19 9:35:00 AM |
|---|---|---|

Font color: Text 1

| Page 15: [28] Formatted | Authors | 8/13/19 9:35:00 AM |
|---|---|---|

Font color: Text 1

| Page 15: [29] Formatted | Authors | 8/13/19 9:35:00 AM |
|---|---|---|

Font color: Text 1

| Page 15: [30] Formatted | Authors | 8/13/19 9:35:00 AM |
|---|---|---|

Font color: Text 1

| Page 15: [31] Formatted | Authors | 8/13/19 9:35:00 AM |
|---|---|---|

Font color: Text 1

| Page 15: [32] Formatted | Authors | 8/13/19 9:35:00 AM |
|---|---|---|

Font color: Text 1

| Page 15: [33] Formatted | Authors | 8/13/19 9:35:00 AM |
|---|---|---|

Font color: Text 1

| Page 15: [34] Formatted | Authors | 8/13/19 9:35:00 AM |
|---|---|---|

Font color: Text 1

| Page 15: [35] Formatted | Authors | 8/13/19 9:35:00 AM |
|---|---|---|

Font color: Text 1

| Page 15: [35] Formatted | Authors | 8/13/19 9:35:00 AM |
|---|---|---|

Font color: Text 1

| Page 16: [36] Formatted | Authors | 8/13/19 9:35:00 AM |
|---|---|---|

Font color: Text 1

| Page 16: [36] Formatted | Authors | 8/13/19 9:35:00 AM |
|---|---|---|

Font color: Text 1

| Page 16: [37] Formatted | Authors | 8/13/19 9:35:00 AM |
|---|---|---|

Font color: Text 1

| Page 16: [38] Formatted | Authors | 8/13/19 9:35:00 AM |
|---|---|---|

Font color: Text 1

| Page 16: [39] Formatted | Authors | 8/13/19 9:35:00 AM |
|---|---|---|

Font color: Text 1

| Page 16: [40] Formatted | Authors | 8/13/19 9:35:00 AM |
|---|---|---|

Font color: Text 1

| Page 16: [41] Formatted | Authors | 8/13/19 9:35:00 AM |

Font color: Text 1

| Page 16: [41] Formatted | Authors | 8/13/19 9:35:00 AM |

Font color: Text 1

| Page 16: [42] Formatted | Authors | 8/13/19 9:35:00 AM |

Font color: Text 1

| Page 16: [43] Formatted | Authors | 8/13/19 9:35:00 AM |

Font color: Text 1

| Page 16: [44] Formatted | Authors | 8/13/19 9:35:00 AM |

Font color: Text 1

| Page 16: [45] Formatted | Authors | 8/13/19 9:35:00 AM |

Font color: Text 1

| Page 16: [46] Formatted | Authors | 8/13/19 9:35:00 AM |

Font color: Text 1

| Page 16: [47] Formatted | Authors | 8/13/19 9:35:00 AM |

Font color: Text 1

| Page 16: [48] Formatted | Authors | 8/13/19 9:35:00 AM |

Font color: Text 1

| Page 16: [49] Formatted | Authors | 8/13/19 9:35:00 AM |

Font color: Text 1

| Page 16: [49] Formatted | Authors | 8/13/19 9:35:00 AM |

Font color: Text 1

| Page 16: [50] Formatted | Authors | 8/13/19 9:35:00 AM |

Font color: Text 1

| Page 16: [50] Formatted | Authors | 8/13/19 9:35:00 AM |

Font color: Text 1

| Page 16: [51] Formatted | Authors | 8/13/19 9:35:00 AM |

Font color: Text 1

| Page 16: [52] Formatted | Authors | 8/13/19 9:35:00 AM |

Font color: Text 1

| Page 16: [53] Formatted | Authors | 8/13/19 9:35:00 AM |

Font color: Text 1

| Page 16: [54] Formatted | Authors | 8/13/19 9:35:00 AM |

Font color: Text 1

| Page 16: [55] Formatted | Authors | 8/13/19 9:35:00 AM |

Font color: Text 1

| Page 16: [55] Formatted | Authors | 8/13/19 9:35:00 AM |

Font color: Text 1

| Page 16: [56] Formatted | Authors | 8/13/19 9:35:00 AM |
|---|---|---|

Font color: Text 1

| Page 16: [57] Formatted | Authors | 8/13/19 9:35:00 AM |
|---|---|---|

Font color: Text 1

| Page 16: [58] Formatted | Authors | 8/13/19 9:35:00 AM |
|---|---|---|

Font color: Text 1

| Page 16: [59] Formatted | Authors | 8/13/19 9:35:00 AM |
|---|---|---|

Font color: Text 1

| Page 16: [60] Formatted | Authors | 8/13/19 9:35:00 AM |
|---|---|---|

Font color: Text 1

| Page 16: [61] Formatted | Authors | 8/13/19 9:35:00 AM |
|---|---|---|

Font color: Text 1

| Page 16: [62] Formatted | Authors | 8/13/19 9:35:00 AM |
|---|---|---|

Font color: Text 1

| Page 16: [63] Deleted | Authors | 8/13/19 9:35:00 AM |
|---|---|---|
| Page 16: [63] Deleted | Authors | 8/13/19 9:35:00 AM |
| Page 16: [63] Deleted | Authors | 8/13/19 9:35:00 AM |
| Page 16: [63] Deleted | Authors | 8/13/19 9:35:00 AM |
| Page 16: [63] Deleted | Authors | 8/13/19 9:35:00 AM |
| Page 16: [63] Deleted | Authors | 8/13/19 9:35:00 AM |
| Page 16: [63] Deleted | Authors | 8/13/19 9:35:00 AM |
| Page 16: [63] Deleted | Authors | 8/13/19 9:35:00 AM |
| Page 16: [63] Deleted | Authors | 8/13/19 9:35:00 AM |

| Page 16: [64] Formatted | Authors | 8/13/19 9:35:00 AM |
|---|---|---|

Font color: Text 1

| Page 16: [65] Formatted | Authors | 8/13/19 9:35:00 AM |
|---|---|---|

Font color: Text 1

| Page 17: [66] Formatted | Authors | 8/13/19 9:35:00 AM |
|---|---|---|

Font color: Text 1

| Page 17: [67] Formatted | Authors | 8/13/19 9:35:00 AM |
|---|---|---|

Font color: Text 1

| Page 17: [67] Formatted | Authors | 8/13/19 9:35:00 AM |
|---|---|---|

Font color: Text 1

| Page 17: [68] Formatted | Authors | 8/13/19 9:35:00 AM |
|---|---|---|

Font color: Text 1

| Page 17: [69] Formatted | Authors | 8/13/19 9:35:00 AM |
|---|---|---|

Font color: Text 1

| Page 17: [70] Formatted | Authors | 8/13/19 9:35:00 AM |
|---|---|---|

Font color: Text 1

| Page 17: [71] Deleted | Authors | 8/13/19 9:35:00 AM |
|---|---|---|

| Page 17: [72] Formatted | Authors | 8/13/19 9:35:00 AM |
|---|---|---|

Font color: Text 1

| Page 17: [73] Formatted | Authors | 8/13/19 9:35:00 AM |
|---|---|---|

Font color: Text 1

| Page 17: [74] Formatted | Authors | 8/13/19 9:35:00 AM |
|---|---|---|

Font color: Text 1

| Page 17: [75] Formatted | Authors | 8/13/19 9:35:00 AM |
|---|---|---|

Font color: Text 1

| Page 17: [76] Formatted | Authors | 8/13/19 9:35:00 AM |
|---|---|---|

Font color: Text 1

| Page 17: [76] Formatted | Authors | 8/13/19 9:35:00 AM |
|---|---|---|

Font color: Text 1

| Page 17: [77] Deleted | Authors | 8/13/19 9:35:00 AM |
|---|---|---|

| Page 17: [77] Deleted | Authors | 8/13/19 9:35:00 AM |
|---|---|---|

| Page 17: [77] Deleted | Authors | 8/13/19 9:35:00 AM |
|---|---|---|

| Page 17: [77] Deleted | Authors | 8/13/19 9:35:00 AM |
|---|---|---|

| Page 17: [77] Deleted | Authors | 8/13/19 9:35:00 AM |
|---|---|---|

| Page 17: [77] Deleted | Authors | 8/13/19 9:35:00 AM |
|---|---|---|

| Page 17: [77] Deleted | Authors | 8/13/19 9:35:00 AM |
|---|---|---|

| Page 17: [77] Deleted | Authors | 8/13/19 9:35:00 AM |
|---|---|---|

| Page 17: [77] Deleted | Authors | 8/13/19 9:35:00 AM |

| Page 17: [77] Deleted | Authors | 8/13/19 9:35:00 AM |

| Page 17: [78] Formatted | Authors | 8/13/19 9:35:00 AM |

Font color: Text 1

| Page 17: [79] Formatted | Authors | 8/13/19 9:35:00 AM |

Font color: Text 1

| Page 17: [80] Formatted | Authors | 8/13/19 9:35:00 AM |

Font color: Text 1

| Page 17: [80] Formatted | Authors | 8/13/19 9:35:00 AM |

Font color: Text 1

| Page 17: [81] Formatted | Authors | 8/13/19 9:35:00 AM |

Font color: Text 1

| Page 17: [82] Formatted | Authors | 8/13/19 9:35:00 AM |

Font color: Text 1

| Page 17: [83] Formatted | Authors | 8/13/19 9:35:00 AM |

Font color: Text 1

| Page 17: [84] Formatted | Authors | 8/13/19 9:35:00 AM |

Font color: Text 1

| Page 17: [85] Formatted | Authors | 8/13/19 9:35:00 AM |

Font color: Text 1

| Page 17: [85] Formatted | Authors | 8/13/19 9:35:00 AM |

Font color: Text 1

| Page 17: [86] Formatted | Authors | 8/13/19 9:35:00 AM |

Font color: Text 1

| Page 17: [87] Formatted | Authors | 8/13/19 9:35:00 AM |

Font color: Text 1

| Page 17: [88] Formatted | Authors | 8/13/19 9:35:00 AM |

Font color: Text 1

| Page 17: [89] Formatted | Authors | 8/13/19 9:35:00 AM |

Font color: Text 1

| Page 17: [90] Formatted | Authors | 8/13/19 9:35:00 AM |

Font color: Text 1

| Page 17: [91] Formatted | Authors | 8/13/19 9:35:00 AM |

Font color: Text 1

| Page 17: [92] Formatted | Authors | 8/13/19 9:35:00 AM |
|---|---|---|

Font color: Text 1

| Page 17: [93] Formatted | Authors | 8/13/19 9:35:00 AM |
|---|---|---|

Font color: Text 1

| Page 17: [94] Formatted | Authors | 8/13/19 9:35:00 AM |
|---|---|---|

Font color: Text 1

| Page 17: [95] Deleted | Authors | 8/13/19 9:35:00 AM |
|---|---|---|

▼

| Page 17: [96] Formatted | Authors | 8/13/19 9:35:00 AM |
|---|---|---|

Font color: Text 1

**Captions of Table and Figures:**

Table 1. Details of the six products evaluated during 2011-2013.

[revised manuscript text omitted]